# Responding to eruptive transitions during the 2020–2021 eruption of La Soufrière volcano, St. Vincent

E. P. Joseph [1✉], M. Camejo-Harry [1,2], T. Christopher[1,3], R. Contreras-Arratia [1], S. Edwards [1], O. Graham [1], M. Johnson[1], A. Juman[1], J. L. Latchman[1], L. Lynch[1], V. L. Miller [1,3], I. Papadopoulos [1], K. Pascal[1,3], R. Robertson [1], G. A. Ryan [1,3], A. Stinton[1,3], R. Grandin[4], I. Hamling [5], M-J. Jo[6], J. Barclay [7], P. Cole [8], B. V. Davies [7] & R. S. J. Sparks[9]

A critical challenge during volcanic emergencies is responding to rapid changes in eruptive behaviour. Actionable advice, essential in times of rising uncertainty, demands the rapid synthesis and communication of multiple datasets with prognoses. The 2020–2021 eruption of La Soufrière volcano exemplifies these challenges: a series of explosions from 9–22 April 2021 was preceded by three months of effusive activity, which commenced with a remarkably low level of detected unrest. Here we show how the development of an evolving conceptual model, and the expression of uncertainties via both elicitation and scenarios associated with this model, were key to anticipating this transition. This not only required input from multiple monitoring datasets but contextualisation via state-of-the-art hazard assessments, and evidence-based knowledge of critical decision-making timescales and community needs. In addition, we share strategies employed as a consequence of constraints on recognising and responding to eruptive transitions in a resource-constrained setting, which may guide similarly challenged volcano observatories worldwide.

[1] Seismic Research Centre, The University of the West Indies, St. Augustine, Trinidad and Tobago. [2] Department of Earth Science, University of Oxford, Oxford OX1 3AN, UK. [3] Montserrat Volcano Observatory, Flemmings, Montserrat. [4] Université de Paris, Institut de Physique du Globe de Paris, CNRS, F-75005 Paris, France. [5] GNS Science, Lower Hutt, New Zealand. [6] USRA, NASA-GSFC, Greenbelt, MD, USA. [7] School of Environmental Sciences, University of East Anglia, Norwich, UK. [8] School of Geography, Earth and Environmental Sciences, Plymouth University, Drake Circus, Plymouth, UK. [9] School of Earth Sciences, University of Bristol, Bristol BS8 1 R J, UK. ✉email: erouscilla.joseph@sta.uwi.edu

A major goal of volcanology is to forecast changes in the behaviour of volcanoes, particularly the onset and conclusion of eruptions and behavioural transitions, such as between explosive and effusive activity[1–3]. Transitions pose challenges for decision-makers in the management of ongoing volcanic crises; >75% of recent fatalities are associated with changing eruptive behaviour, where all or some individuals were inside declared hazard zones at that point[4]. The 2020–2021 La Soufrière volcanic eruption in St. Vincent, illustrates these challenges.

Crisis science is defined as conducting scientific research during a crisis, which involves data acquisition, analysis, interpretation and archiving of scientific and technical resources, as well as organising logistics, staffing and communicating findings with stakeholders and the public[5]. Volcanic crises include unrest, eruption and any aftermath. The core goal of observatory staff during crises is to acquire, analyse interpret and communicate data in a way that assists local populations and civil protection agencies with their decision-making[6].

La Soufrière volcano (13.33°N; 61.18°W), located in northern St. Vincent, is a 1220 m high stratovolcano with a summit crater ~1.6 km in diameter and 300–600 m deep[7]. Historical eruptions of basaltic-andesitic magmas typically last many months[8,9] and occur in both the absence and presence of a crater lake. Eruptions have been both explosive (1718[7], 1812, 1902–03, 1979) and effusive (1784, 1971–72). The most recent 1979 eruption ended with a 120 m × 860 m lava dome emplaced in the crater. As in many volcanoes worldwide, fatalities are associated with rapid accelerations in explosive activity[4] exemplified by the 1812 and 1902–03 St. Vincent eruptions.

Precursory unrest was markedly low-level prior to the 1971–1972 and 1979 eruptions, which began with <24 h of instrumentally recorded precursory seismicity[9]. Since the 1800s, several episodes of unrest without eruption (crater lake temperature changes, felt seismicity) have also occurred.

Increased background seismicity at La Soufrière from 1 November into December 2020 prompted an inspection of the crater by staff of the Soufrière Monitoring Unit (SMU), of St. Vincent and the Grenadines (SVG) National Emergency Management Organisation (NEMO), on 16 November 2020. Minor changes in fumarolic activity on the dome and the small lake occupying the eastern crater floor were noted. Seismicity reduced after 23 December 2020.

Surface activity was first recognised on 27 December 2020, when the NASA Fire Information for Resource Management System (FIRMS) detected a thermal anomaly inside the summit crater. On 29 December 2020, thermal anomalies and greyish-white emissions were observed. SMU staff discovered a new dome located in the west south-west sector of the crater, adjacent to the 1979 dome. Effusive activity continued for three months, with a rapid increase in effusion rate in early April 2021 leading to the explosive phase between 9 and 22 April. Thereafter, activity was limited to moderate $SO_2$ outgassing and generally low-level seismicity.

Here we describe monitoring data and the evolution of scientific interpretations. We also reflect on the information most critical to the generation of actionable forecasts of eruptive transition in a 'real world' setting.

## Results and discussion

**Network strengthening.** Overall emergency management and scientific support for La Soufrière volcano were coordinated by the staff of The University of the West Indies Seismic Research Centre (UWI SRC) in Trinidad, with assistance from the Montserrat Volcano Observatory (MVO) as well as regional and international collaborating partners. Initial observations and local support were provided by the SMU.

Limited resources and the COVID pandemic resulted in a much-reduced monitoring capacity at the onset of unrest (November 2020), with one working seismic station (SVB) 9 km from the volcano, and one continuous GPS station (SVGB) (Fig. 1). SRC reactivated the local observatory in late December 2020 and upgraded the monitoring network (Fig. 1). Eight broadband seismic stations were operating by the end of February 2021 and the ground deformation network was augmented by four continuous GPS sites (SVGR, SVGS, SVGF, SVGG) in addition to re-occupation of two campaign benchmark sites (JCWL and TBRK) (Fig. 1). A 9-prism EDM target was installed on the southern crater rim and weekly measurements attempted from six locations. Interferometric Synthetic Aperture Radar (InSAR) processing of available ALOS-2 and Sentinel-1 images augmented ground deformation monitoring. Sentinel-2 and PlanetLabs satellite imaged both the crater and colour changes of vegetation on the volcano's flanks. Cameras installed at the Belmont Observatory (3 January 2021) and crater rim (24 January 2021), multispectral and radar satellite imagery, oblique aerial and terrestrial photographs and UAV aerial photography and photogrammetry allowed visual observations to document dome growth. From 14 January 2021, gas emissions were measured using a Multi-component Gas Analysing System (MultiGAS) and Ultra-Violet (UV) spectroscopy.

**Seismic monitoring.** Seismicity increased slightly on station SVB in November 2020, but remained modest until 23 December 2020, averaging two events per day, with a maximum magnitude of 3.3 Mt and no reported felt events (Fig. 2).

Although dome extrusion started on 27 December 2020, no seismicity was recorded until 6 January 2021, with an average of two events per day up to 17 January, when there was a sharp increase to 60 events per day. Subsequently, low frequency (0.5–5 Hz)[10,11] events were observed and interpreted as related to the dome emplacement; the events were recorded only by the closest stations indicating a shallow source. Volcano-tectonic (VT) swarms occurred during 23–24 March 2021 (226 events) with >95% located at depths shallower than 5 km; and 5–6 April 2021 (476 events) with an abrupt transition to deeper locations (Fig. 3), which was interpreted as a new volume of magma ascending from ~10 km depth.

Banded tremor[12], of increasing magnitude, began around noon (UTC time) on 8 April 2021, recorded by the closest stations at intervals of ~2.5 h (Fig. 4a). This change was interpreted as indicating an imminent explosive phase[13], with a source attributed to the excitation of shallow gas and fluid pockets[14,15]. The spectral content up to 10 Hz suggested that the banded tremor consisted of merging VT events (Fig. 4a). The 8th cycle transitioned to continuous tremor with increasing amplitude and stable frequency content over time (Fig. 4), suggesting repetitive events at a constant rate[13]. The first explosion was recorded at 12:41 UTC on 9 April, followed by a period of sustained, but pulsing, explosive activity and tremor from 16:00 UTC on 9 April to 06:00 UTC on 10 April (Fig. 4).

The time series, RSAM and spectrograms of the explosive phase is shown in Fig. 4. The initially rapid rate of explosions and associated tremor made individual seismic events difficult to identify. Four stations stopped transmitting data during the first 36 h of the explosion sequence. Spectrograms for the explosive phase (Fig. 4a) show a larger amplitude but the same stable frequency as during the build-up. Each explosion lasted between 3–23 min, followed by 2–3 h of exponential decay in tremor amplitude (green arrows in Fig. 4a). Over the following two

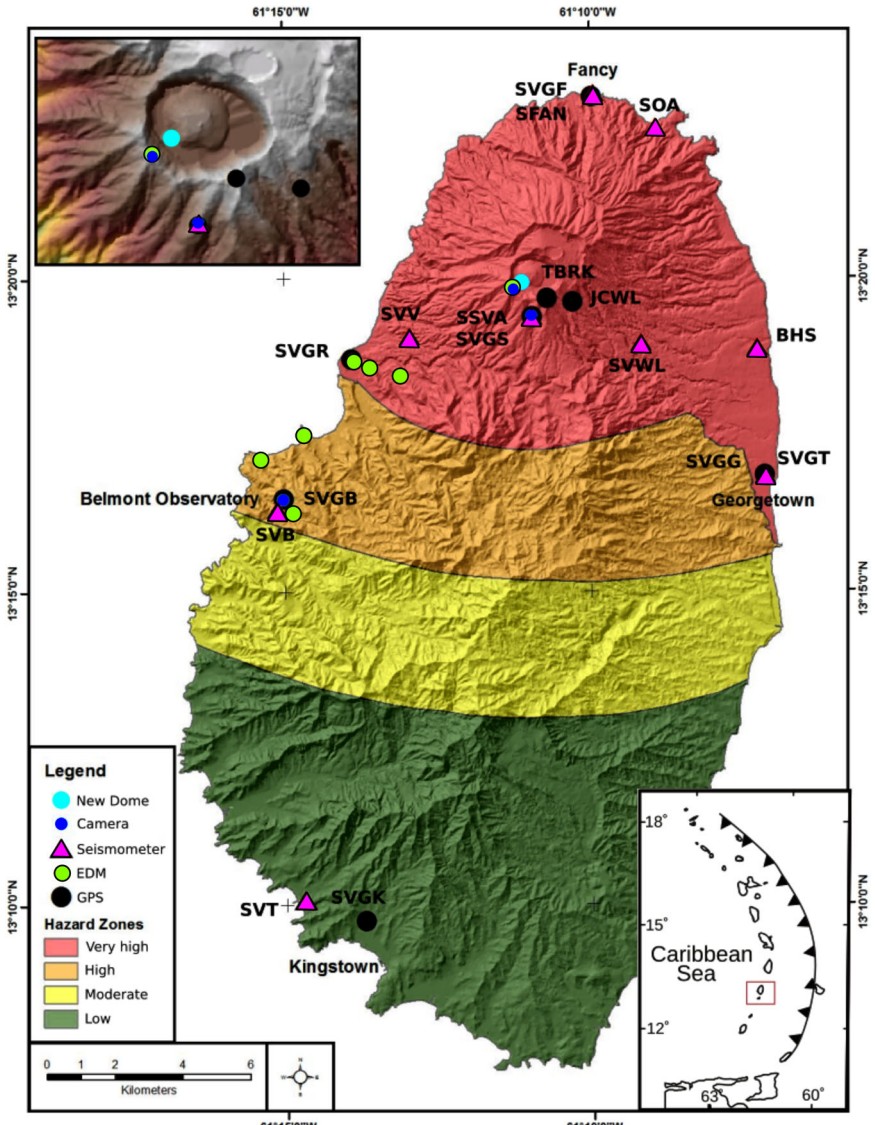

**Fig. 1 Volcano monitoring network for La Soufriere volcano, St. Vincent.** Hazard zones illustrate the potential for ground-based volcanic impacts such as pyroclastic flows and surges, tephra fall, ash fall and lahars that may impact the defined areas[7]. The boundaries of the zones are based on the past incidence of the hazards and areas of maximum projected extent, in addition, experience of these hazards at similar volcanoes is combined with theoretical considerations of mass discharge rates of magma, wind direction and morphology. The effects of effusive eruptions have had little impact on the determination of hazard zones.

weeks, the pattern of seismic activity included episodes of short tremor bands accompanied by enhanced venting or explosive activity. Episodes of tremor were interspersed with long-period and hybrid earthquakes, with their rates of occurrence gradually decreasing prior to a period of high-level tremor on 22 April. The last explosion on 22 April was preceded by several hours of increasing-amplitude tremor (Fig. 4b), with an abrupt end of low-frequency tremor shortly after the explosion. Seismic activity steadily declined from 22 April to early May, from an average of 354 events/day to 24 events/day. Up to November 2021, the seismicity remained sparse, dominated by low-frequency events.

**Ground deformation monitoring.** Forewarning of the effusive eruption was not recognised on the existing continuously operating GPS network (Fig. 2). However, a <10 cm line-of-sight shortening signal was observed in the crater area using ALOS-2 and Sentinel-1 radar, sometime between 19 and 31 December

2020. The associated deformation source was modelled as a ~63,000 m$^3$ dike intrusion, shallower than ~500 m deep (Fig. 5). Subsequently, no deformation was detected from the SAR platforms. No deformation was detected on the EDM time series during the effusive phase.

Onset of the explosive phase was accompanied by a rapid deflation recorded on the continuous GPS network on 9 April 2021 (Fig. 2). Between 9 and 22 April 2021, the SVGB station (Fig. 2) measured an overall cumulative horizontal displacement of ~43 mm northward and ~37 mm eastward and a subsidence of ~81 mm. Using a Mogi point source[16], the associated surface deformation was modelled by migration of ~50 × 10$^6$ m$^3$ of magma from a source at ~6 km depth. After explosive activity ended, slow deflation was observed over several months.

**Gas and geochemical monitoring.** Gas measurements in January 2021, using UV spectrometer and MultiGAS instruments, detected no SO$_2$. The concentration ratio (ppm) of carbon to total

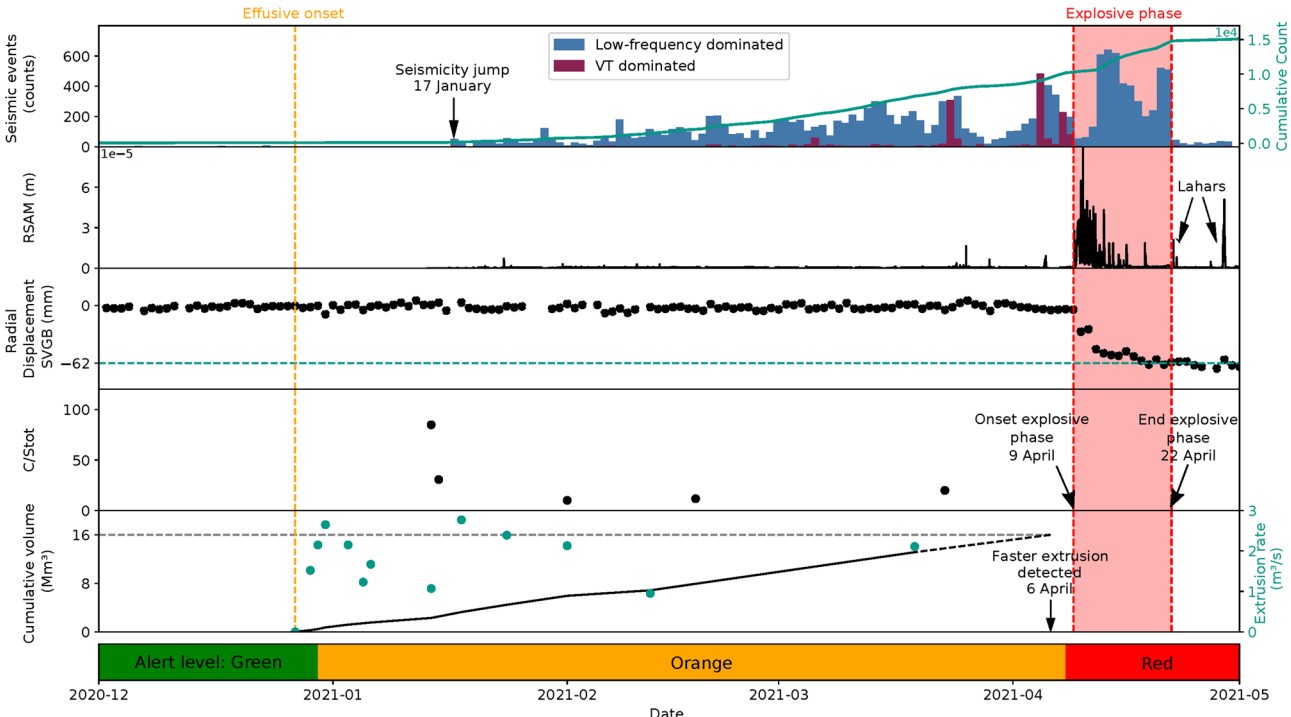

**Fig. 2 Timeline of monitoring data for the 2020–2021 eruption at La Soufrière volcano, St. Vincent.** Seismicity data: daily (bars) and cumulative (teal line) seismicity observed and a marker on the first important increase (17 January). The bars in blue represent seismic events related to fluid dynamics (low-frequency and dome emplacement) and the bars in dark red represent VT events. RSAM values: calculated with 1 min windows and no overlap; they correlate with VT swarms and explosive phase. Moreover, it shows evidence of lahar signals after the explosive phase. Deformation: radial extension from the vent observed at station SVGB, 9 km away from the crater, and associated uncertainties computed with GAMIT/GLOBK[48]. It shows a total movement of 62 mm towards the crater at the end of the explosive phase. $C/S_{tot}$ ($CO_2/H_2S$) concentration ratios (ppm) in the plume from MultiGAS measurements: First two data points evidence only $H_2S$ content, the remaining are a combination of $H_2S$ and $SO_2$. The arrow shows the onset of explosive activity. Dome extrusion data: cumulative volume extruded in black line with an extrapolation until 6 April, extrusion rate in teal dots; the arrow marks the onset of rapid dome inflation as observed by a remote camera. The lowest bar shows the corresponding alert level for each day. In addition, vertical dashed lines show the onset of the effusive phase (orange) and the red area corresponds to the explosive phase.

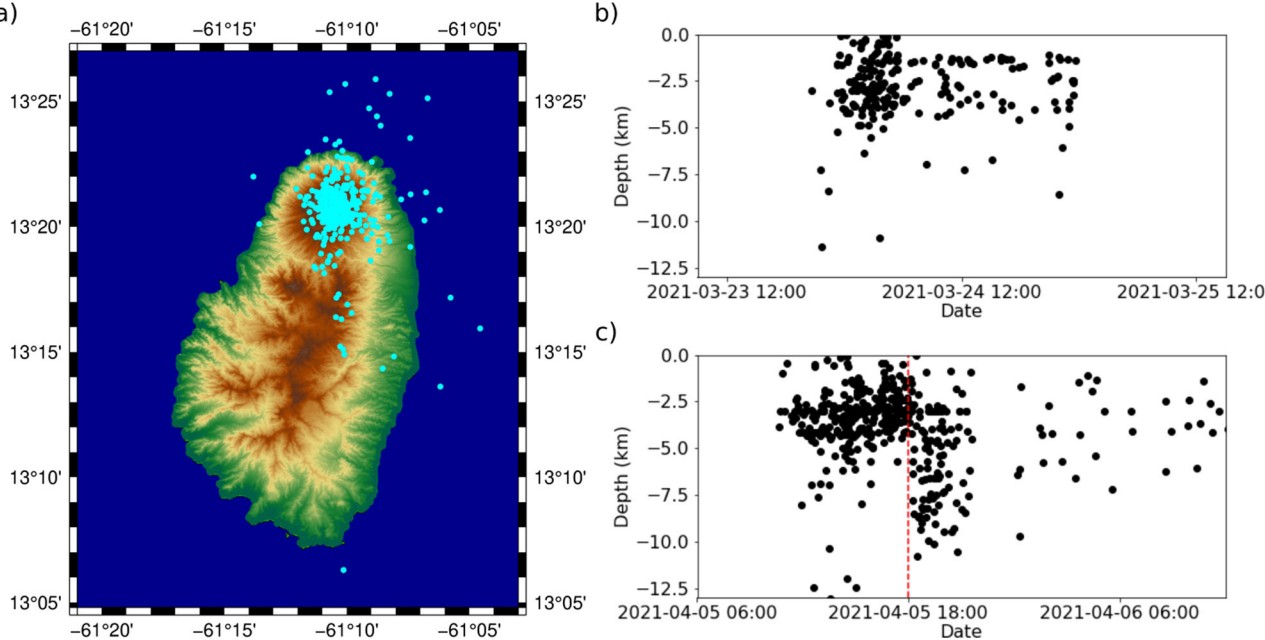

**Fig. 3 Seismicity located during 23–24 March and 5–6 April VT swarms. a** Epicentres calculated showing concentric distribution. **b** Temporal evolution of depths during 23–24 March, most seismicity is shallower than 5 km. **c** Temporal evolution of depths during 5–6 April, seismicity is shallower than 5 km until 18:00 UTC on 5 April, when locations transitioned to deeper levels.

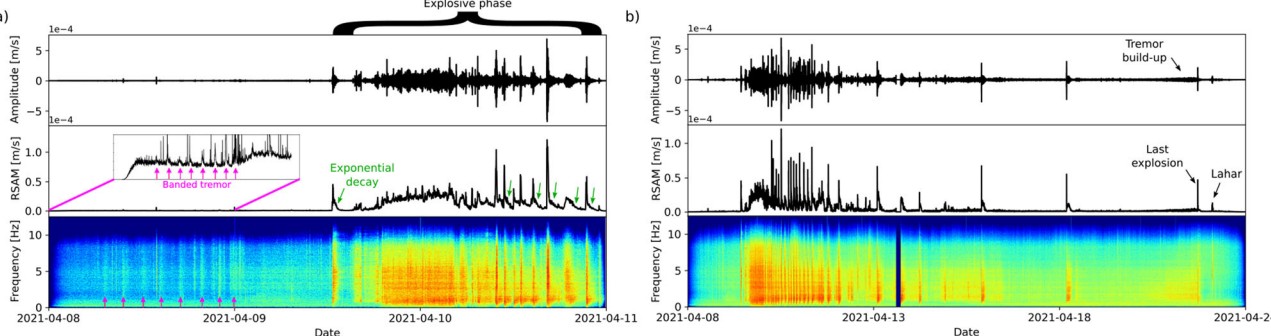

**Fig. 4 Main features of seismicity during explosive phase. a** Vertical time series at station SVV, RSAM and spectrogram showing the main features of the signal during the first 3 days: banded tremor, continuous and discrete explosions with exponential decays. **b** Vertical time series at station SVV, RSAM and spectrogram showing the main features of the whole explosive phase: increasing inter-explosion time, tremor build up to the last explosion and an abrupt end of the low-frequency tremor two hours after the referenced explosion. It also shows a lahar signal around 8 h after.

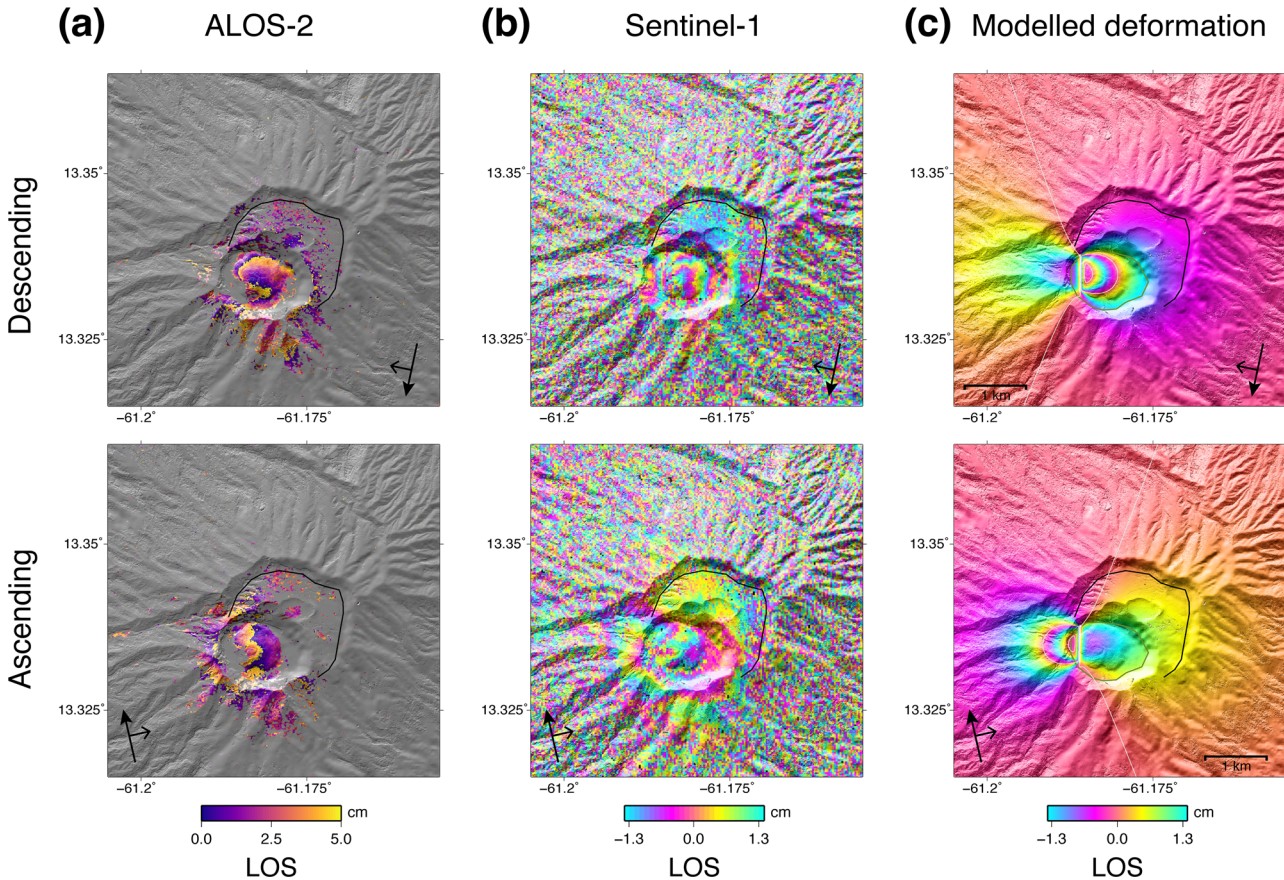

**Fig. 5 Deformation associated with the 27 December 2020 extrusive eruption.** These visualisations use the Grandin and Delorme (2021) DSM.
**a** Interferograms for descending (2020/02/25–2021/01/26, path 131, frame 3350) and ascending (2020/01/15–2021/01/13, path 36, frame 250) ALOS-2 radar. One fringe represents 11.4 cm in the line of sight; **b** Interferograms for descending pass (2020/12/07–2020/12/31, track 156, incidence 44.5°) and ascending (2020/12/07–2020/12/31, track 164, incidence 43.9°) Sentinel-1 radar. One fringe represents 2.8 cm in the line of sight; **c** Synthetic deformation predicted by an Okada dislocation dipping vertically (along-strike length: 600 m, along-dip width: 700 m, upper edge depth: 15 m), opening with a potency of 63,000 m$^3$.

sulphur (C/S$_t$) was measured by MultiGAS at the summit (Fig. 2). A C/S$_t$ ($=CO_2/H_2S$) ratio of 85 and 30.6 (Fig. 2) was obtained on 14 and 15 January, respectively. Minor $SO_2$ (<1 ppm) was detected in February 2021, with a C/S$_t$ ($=SO_2 + H_2S$) concentration ratio of 10 measured on 1 February and 11 on 18 February, before increasing to ~20 on 23 March 2021. Plume compositions during the effusive phase were dominated by a hydrothermal signature (Fig. 6). On the afternoon of 8 April, a

coastal traverse yielded the first detection of $SO_2$ in the gas plume with a mass flux of 80 tonnes/day. The TROPOMI instrument on board Sentinel-5P also detected $SO_2$ during an overpass on 8 April at 17:25 UTC, confirming the change in plume composition. During the explosive phase (9–22 April) only satellite (Sentinel-5P) $SO_2$ measurements were possible, with values ranging from $2.76 \times 10^5$ tonnes/day on 10 April to 331 tonnes/day on 22 April. Over the two months following the explosive phase,

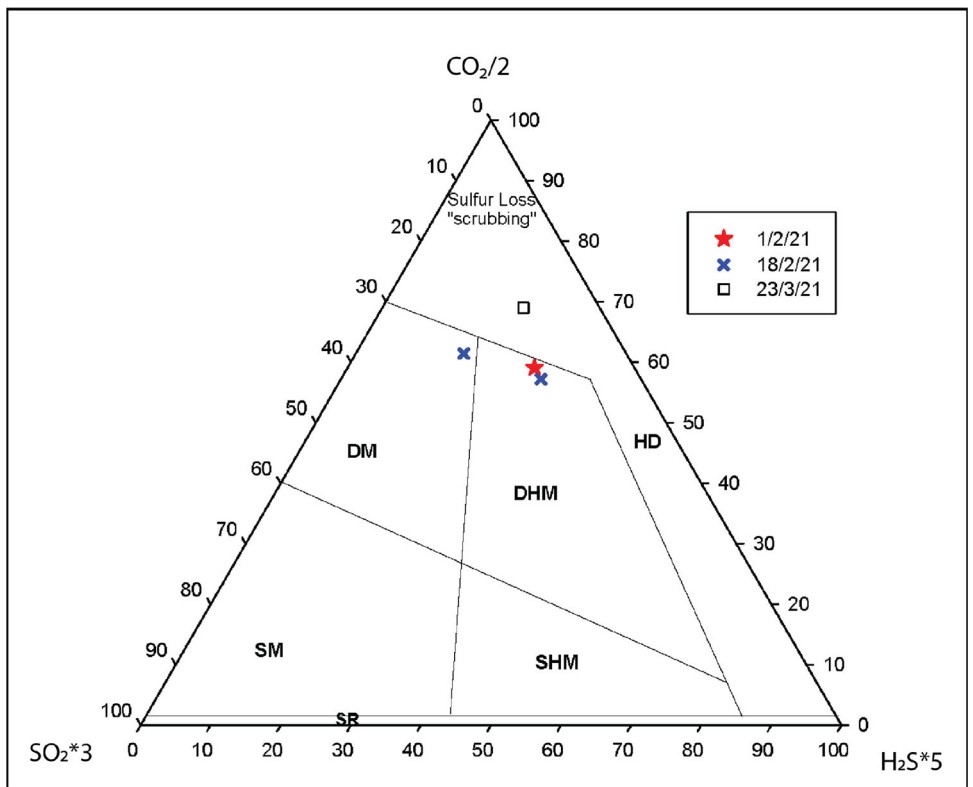

**Fig. 6 Plume composition during the extrusive phase from MultiGAS measurements.** Ternary diagram of $SO_2$*3-$CO_2$/2-$H_2S$*5 showing the plume compositions obtained during the extrusive phase. HD hydrothermal dominated, DHM deep hydrothermal magmatic, SHM shallow hydrothermal magmatic, DM deep magmatic, SM shallow magmatic. Boundaries were obtained from the Central American Volcanic arc.

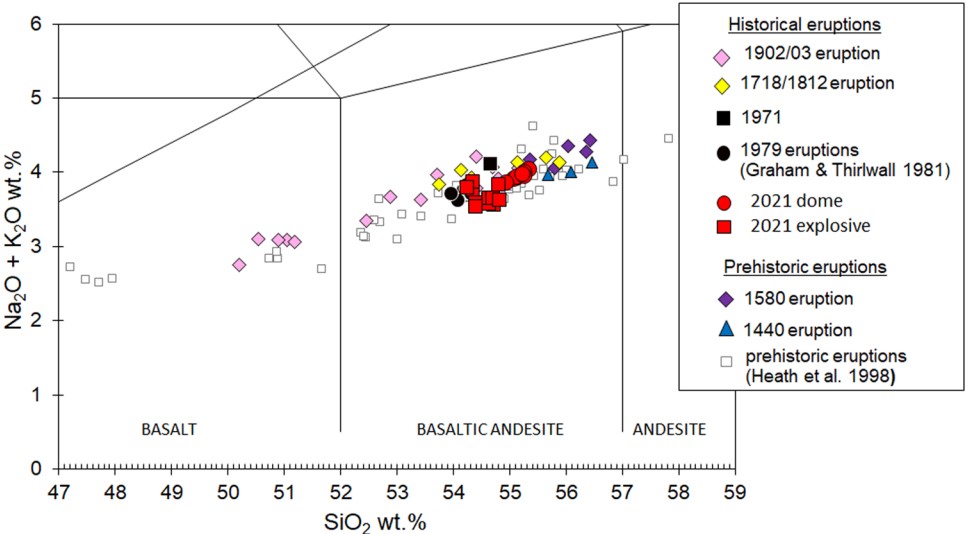

**Fig. 7 Composition of the 2020–2021 dome and other explosive products.** TAS classification diagram ('Total Alkalis vs Silica') to compare the composition of the 2020–2021 dome and other explosive products with other eruptions of La Soufrière, St. Vincent. Results were obtained by XRF at the University of Plymouth.

coastal traverse measurements of $SO_2$ flux decreased from ~800 to ~200 tonnes/day, and then maintained this average through to November 2021.

On 16 January 2021, samples were collected from the front of the lava dome. In April 2021, scoria and clasts from pyroclastic density currents emplaced during the explosive eruptions were sampled. XRF analysis of major elements shows both have basaltic andesite bulk compositions (Fig. 7).

Preliminary petrographic analyses of the dome rocks indicated a phenocryst assemblage similar to past eruptions, consisting of plagioclase, clinopyroxene and Fe-Ti oxides with sparse olivine and abundant gabbroic clots[17] (Fig. 8a, b). Where present, olivine is invariably heavily altered, with symplectites intergrowth of forsterite and Fe-oxides common (Fig. 8b). Groundmass textures show evidence of late-stage disequilibrium, including groundmass crystals with localised alteration of orthopyroxene microlites to Fe

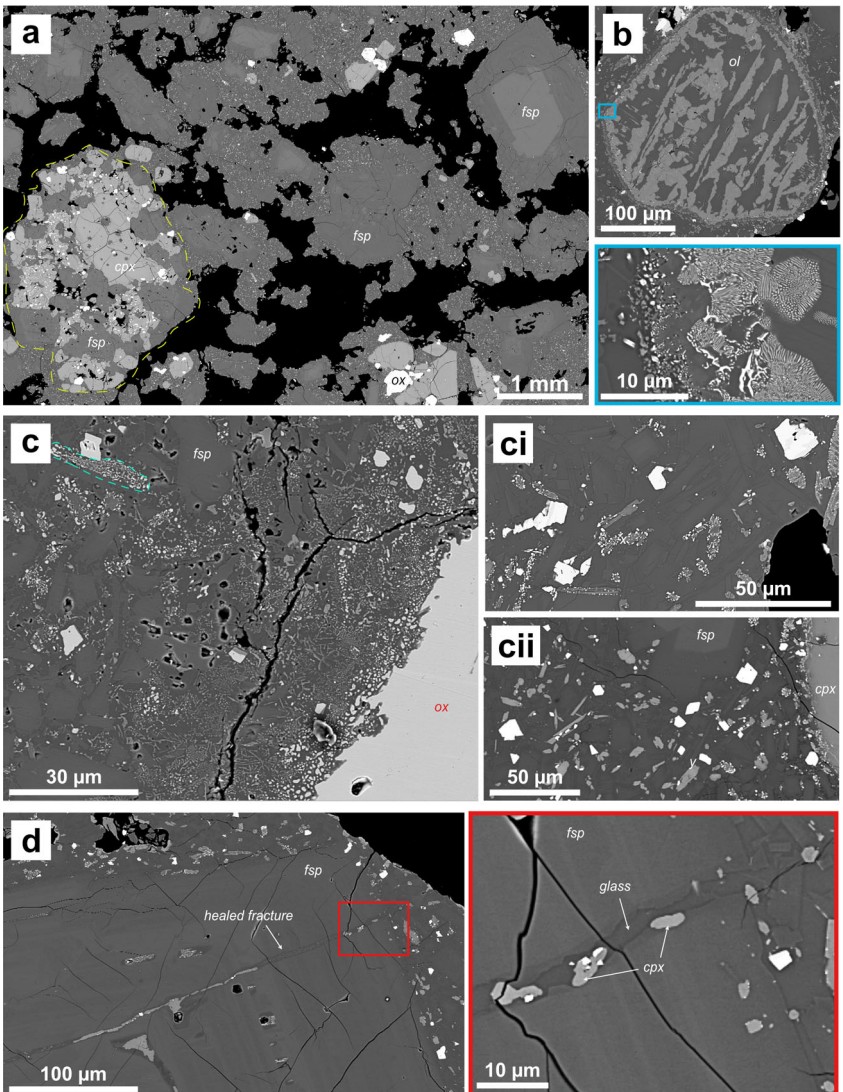

**Fig. 8 Backscattered electron images of the 2021 lava dome.** Five full thin section image maps, acquired on a Zeiss Gemini 300 SEM at the University of East Anglia, UK, were generated from clasts sampled in January 2021 from the growing dome and analysed to identify common textures as displayed in **a**–**d**. **a** Image showing vesicular nature of the dome sample (vesicles in black) with examples of the crystal population labelled: plagioclase (fsp), clinopyroxene (cpx), Fe-Ti oxides (ox) and gabbroic clots (outlined by yellow dashed line). **b** Example of heavily degraded olivine (ol) (blue box) shows close-up of this texture. **c** Examples of localisation of degradation of orthopyroxene crystals to oxides (dashed turquoise line in **c**). Higher levels of degradation observed close to larger oxides (e.g. image c and ci, x) comprises ~5% of each thin section. Elsewhere, orthopyroxenes are un-degraded (cii, y). **d** Example of a plagioclase phenocryst (fsp) with a fracture that has been later filled by melt and crystals, this is observed several times across the five sections analysed. Red box shows a close-up of this structure showing clinopyroxene, oxides and glass infilling the fracture.

and Mg oxides (Fig. 8c). Dome rocks are vesicular with textural evidence for fracturing and annealing of fluid pathways (Fig. 8a, d).

**Dome growth and other visual observations.** Initially, the new dome grew uniformly in all directions, reaching 70 m in height, subsequently elongating in the NW-SE direction (Fig. 9). Gas vented through a small depression in the dome's summit. The shape evolved to an elliptical lava coulee with two distinct lobes, confined within the moat between the 1979 dome and the inner wall of the Summit Crater (Fig. 9). Rock fall activity from the margins was very limited, while no deformation of the crater floor was observed in flow fronts. Distinct marginal levees developed with radial and linear flow patterns appearing on the lava surface. Thermal images on 16 January 2021 yielded surface temperatures of up to 600 °C.

Extrusion rates calculated for periods from 1 to 34 days varied between 0.95 and 2.65 m³/s ± 0.59 m³/s with a long-term average of ~1.85 ± 0.14 m³/s (Fig. 2). The cumulative volume reached ~13 × 10⁶ m³ by 19 March 2021, when the dome measured 912 m long, 243 m wide and 105 m high. Extrapolating the linear trend through to 9 April suggests a final volume of ~18 × 10⁶ m³. On 6 April, observation via the installed camera indicated a rapid increase in dome height with incandescence becoming visible over the crater rim from the Belmont Observatory (Fig. 1) on the evening of 8 April. Cyclic gas emissions at the central vent occurred correlated with the banded tremor. During the explosive activity, the new lava dome and significant parts of the 1979 lava dome were destroyed, as confirmed by satellite imagery on 10 April from ICEYE (02:02 UTC) and Capella (14:03 UTC).

As explosive activity intensified close observation of discrete events became more difficult. By 12 April pyroclastic density currents (PDCs) had descended several valleys on the southern

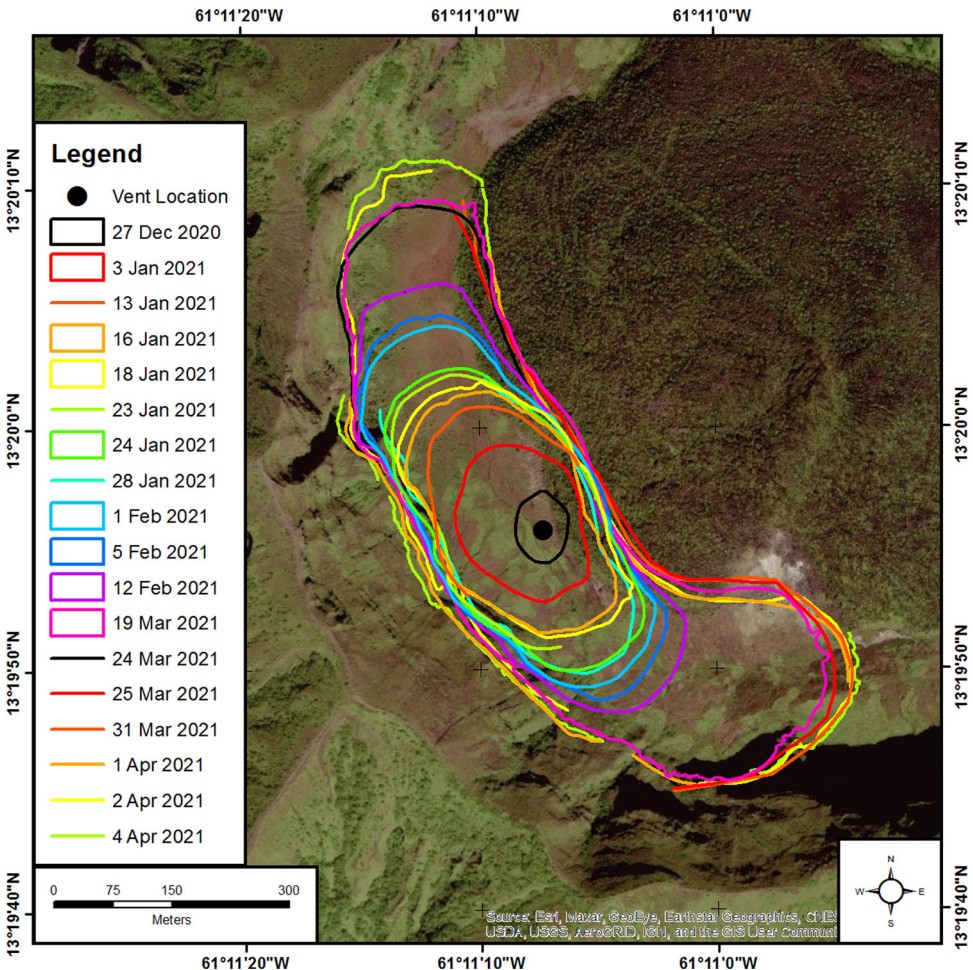

**Fig. 9 Footprints of the 2020–2021 lava dome of La Soufrière volcano, St. Vincent.** A map of the summit crater of La Soufriere, St Vincent, showing full and partial footprints for the new lava dome that first appeared on 27 December 2020. Footprints were extracted from multispectral and radar satellite imagery, oblique aerial and terrestrial photographs and drone surveys. Background is pre-eruption imagery that shows the 1979 lava dome inside the summit crater.

and western flanks of the volcano and reached the sea. Following that, enhanced venting or Vulcanian-style explosive activity episodically occurred until 22 April. Some explosions generated PDCs in valleys on the western flanks of the volcano.

**Crisis response: warning and decision-making systems**. The SRC supports local authorities for strengthening preparedness and communicating volcanic hazards through product development (e.g., integrated volcanic hazard maps). The map for St. Vincent (Fig. 1) is a colour-coded depiction of the expected impact of volcanic hazards across the island[7]. The volcanic alert level system (VALS, see Supplementary Information) translates the volcanic activity level into required actions during volcanic unrest. Operational constraints in country meant that any increased likelihood of an explosive eruption needed to be communicated 24–48 h before onset to enable successful evacuation.

Distinguishing rapid accelerations in activity after the onset of an eruption, in particular, transitions to an explosive eruption, are a globally recognised forecasting challenge[18]. It is also important to recognise when the probability of explosions decreases, to lower the alert level. Here we identify the practical challenges in recognising and communicating these changes. We highlight the importance of preparedness, diverse forms of communication, and structured approaches to the interpretation of scientific data,

development of evidence-informed forecasts and assessment of risk during volcanic crises.

**Short-term contribution to decision-making: assessment of risk**. Current uncertainty in understanding volcanic processes contribute to a variety of opinions on causative mechanisms and prognoses, particularly during an evolving crisis. In addition, the aleatoric uncertainty associated with the complex behaviour of volcanic systems requires caution against a deterministic interpretation that over-emphasises one specific outcome[19]. SRC used the framework of a structured expert elicitation[20,21] around a range of scenarios, to generate both consensus (a collective 'most likely' prognosis) and to represent the diversity of opinions.

Weekly elicitations (January to early March 2021) favoured continuation of effusive activity (~80%) each time. The likelihood of an escalation to explosive activity in the following weeks remained at median probability of ~10%. However, following the first VT swarm (23–24 March), elicited estimates for a transition to explosive activity doubled to a median probability of ~20%. With the appearance of banded tremor (8 April), elicited probabilities of explosive activity tripled to a median value of ~60%. The authorities of SVG were alerted to this increase in volcanic activity. The alert level was raised to Red on 8 April at 18:00 UTC triggering the evacuation of ~16,000 persons from the Red and Orange Zones, prior to the start of explosive activity

on 9 April at 12:41 UTC, with no reported serious injuries or loss of life.

The visual observations of declining surface activity, lowered seismic activity and declining gas output, coupled with the slow deflation signal observed since 22 April, were key drivers for the lowering of the alert level to Orange on 6 May.

**Longer-term contribution to decision-making: risk awareness, preparedness and communication**. Hazard assessments and analysis of past events have continuously been updated in response to new understanding[22–26]. Further, improvements in communication of improved understanding of hazards have been assessed and implemented in hazard planning by NEMO and SRC. The Volcano Ready Community Project (VRCP) led by SRC in collaboration with NEMO, launched in April 2018 and completed in April 2021, targeted twelve northernmost communities of St. Vincent in the Red and Orange hazard zones of the most recent volcanic hazard map[7]. The VRCP, enabled community plans to be drafted and integrated into the national response mechanisms prior to the 2020–2021 eruption.

**Communication pathways: transition from extrusive to explosive**. Communication of messaging between SRC and NEMO was harmonised. A continuous flow of near real-time information was provided to the public and stakeholders about volcanic activity, hazards, and risk reduction. These communications maintained credibility in the monitoring capability of SRC[27,28]. A similar communication strategy employed by the USGS, in response to the 2014–2015 Kilauea volcano lava-flow crisis, was shown to be a highly effective approach[29] and aligns with volcano observatory best practices for operations during crises[6].

Based on best practice and evidence, risk communication products were developed to target different learning styles, media platforms and preferences[28]. These products included visual, print and audio products, and were combined with live scientific presentations during media interviews and to special interest groups. Social media schedules and posts were coordinated, while SRC scientists on island participated in daily activity updates on local television and radio stations, and provided cabinet briefings and updates to decision-makers. Scientists participated in virtual and drive-through community meetings for Red Zone residents with live online streaming and simulcast on local television and radio. An important facet of uncertainty during eruptions is dealing with misinformation and rumours. The strategy of maintaining a continuous presence on social media (Fig. 10) and the use of FAQs and short interviews allowed growing concerns or misapprehensions to be addressed. The frequency of these communications was influenced by changes in the ongoing activity. International scientists were also encouraged to amplify existing messages and use SRC materials in discussing the eruption with their local media.

Another important dimension was systematised internal communication. External contributions of data were facilitated by a team lead who was responsible for internal communication and coordinating data requests. This approach also facilitated international collaborations and engagement with academic scientists, which supported the SRC to develop conceptual models.

With the start of the explosive phase of the eruption, social media posts were still the primary tool used by SRC to communicate with the public. Scientific bulletins were shared directly on these platforms, with the addition of voice notes shared via mobile networks. Daily activity updates on local radio and television stations continued. Visual, print and audio products now also focussed on explanations of, and recommended responses to, the primary volcanic hazards (pyroclastic density currents, ash fall and lahars) being observed.

**Crisis management**. Volcano monitoring data enable scientists to provide short-term forecasts or advise of possible changes during an ongoing eruption[30]. However, for successful crisis management, monitoring data and interpretations need to be: (a) framed within the context of wider scientific knowledge, (b) presented in the context of decision-making ('useful, usable and used')[31] and (c) effectively communicated to diverse audiences[32]. The St. Vincent case provides an important demonstration of how these principles were integrated, complementing synoptic analyses of the state-of-the-art in volcano observatory crisis operations[6,33]. Next, we discuss the key lessons from our analysis of response to the unfolding events, particularly the eruptive transition, and assess the role data and models played in decision-making. We also reflect on the constraints on best practice imposed by finite resources, and how this can be improved.

**Conceptual models and their value in forecasting eruptive transitions**. Historically, La Soufrière volcano can produce both explosive and effusive eruptions over time intervals of weeks to months. However, transitions in behaviour can occur over only a few hours[6] and pose acute challenges to risk management; particularly when decisions to evacuate are exacerbated by resource or space constraints that affect the tolerability of evacuations. Analysis of previous eruptions in St. Vincent has demonstrated that compliance with long-duration evacuations will dissipate, a feature shared with crises at other volcanoes[4].

A working conceptual model of volcanic behaviour was created and developed in real-time, which was used to inform the scientific response to emergency management and advise the authorities. Critically, during the effusive phase our evolving working model was used to anticipate explosive transition or other significant changes in activity.

In early January 2021, we interpreted onset of the eruption as the consequence of the injection of fresh gas-rich magma into a sub-volcanic reservoir, making its way to the surface[3]. However, this interpretation could not explain the comparatively low seismicity rates, lack of surface deformation and near-constant extrusion of lava (Fig. 2). The presence of a ductile well-connected magma ascent pathway was proposed to reconcile these early seismic observations[34]. The absence of deformation and steady extrusion could be explained by either (i) a magma source that maintains a near-constant overpressure[35] or (ii) that a large magmatic source, relative to the material extruded, resulted in pressure decay in the reservoir being too small to be detected[36,37] or (iii) that hot magma mush surrounding the source region, in combination with the viscous flow in the crust, maintained the high pressure[37], or (iv) some combination of these processes. The diversity of explanations, informed by monitoring observations, was important for assessing the potential for explosive activity, and the timescale over which this might happen. At that time, there were no evacuations in place, but existing hazard assessment and outcomes from past simulations (e.g., Tradewinds Exercise 2019[38]) demonstrated that risk to the northern population could rapidly become high, with a 24–48 h interval needed for full evacuation.

By early February, the absence of detectable $SO_2$, however, led us to infer the presence of degassed magma remaining within the conduit, following the 1979 eruption, confined by a strong 'cap', was slowly being pushed up by a new injection of gas-rich magma. By early March, the similar composition and petrographic characteristics of the extruded dome rocks, in comparison to past eruptive products, had reinforced this model. Similar

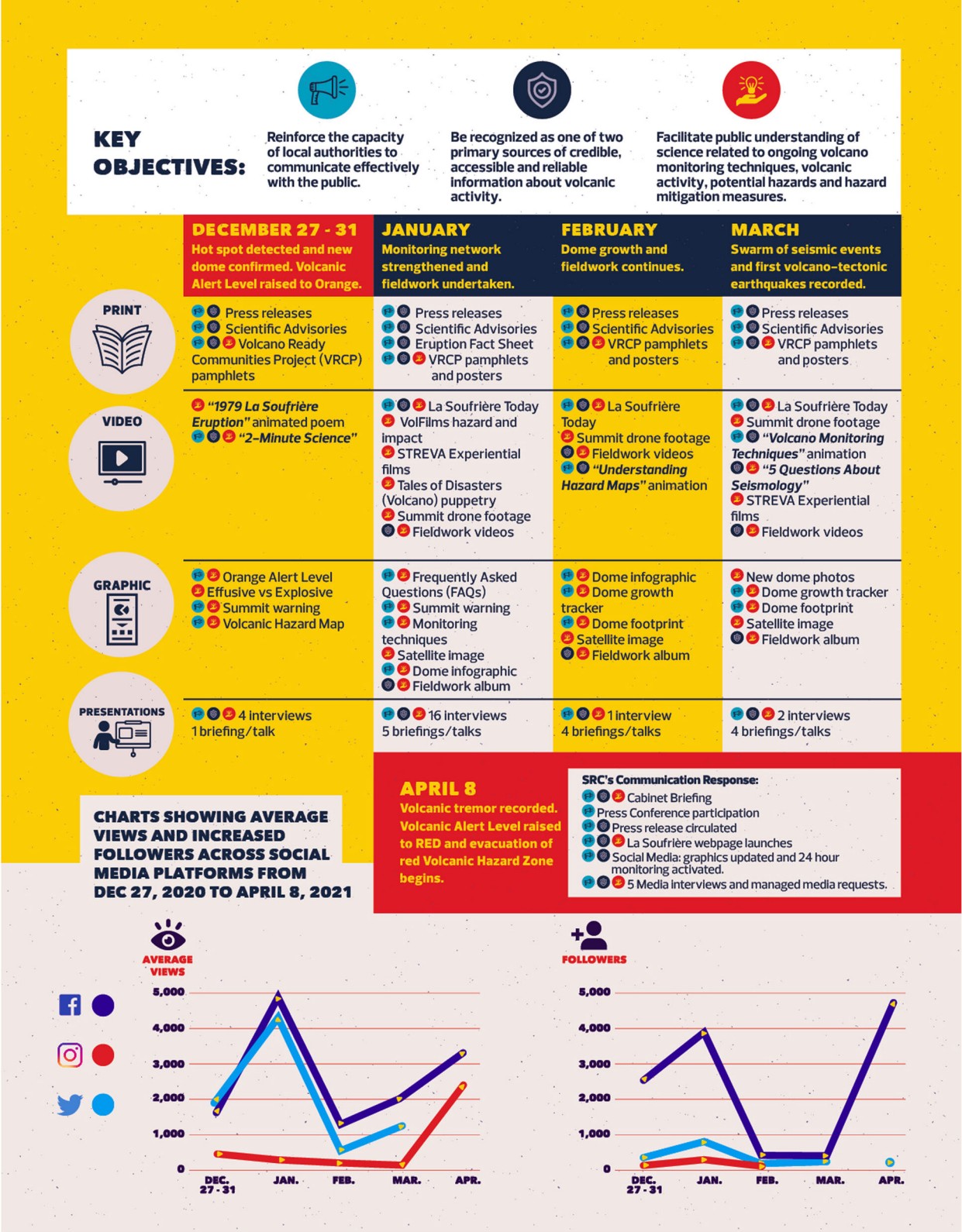

**Fig. 10 UWI SRC communication and eruption response strategy.** Summary of UWI SRC communication strategy and response throughout the 2020–2021 eruption of La Soufrière, St. Vincent eruption.

behaviour has been inferred at volcanoes such as Kilauea in Hawai'i, where in 2018 near real-time geochemical analysis of lava indicated magma characteristics consistent with progressive flushing of residual magma in the conduit[39].

The epicentres of the intense seismic swarms before the explosions show a concentric distribution of earthquakes around the volcano (Fig. 3a). This suggested magma ascending through the volcanic conduit. Most (>95%) of the estimated epicentres

were located above 5 km until 18:00 UTC on 5 April, when seismicity migrated to deeper levels (Fig. 3a, b). This sudden transition in depth was considered evidence of increased deviatoric stresses around the conduit, possibly related to a new intrusion of gas-rich magma.

In our evolving model, obstruction by the cap material and overlying 1979 dome prevented fresh magma from reaching the surface and limited $SO_2$ flux to volumes low enough to be scrubbed by the volcanic hydrothermal system, until April 2021. We speculated that the accelerated extrusion rate observed after 6 April, was after the high-viscosity magma cap was displaced by new lower viscosity gas-rich magma. Banded tremor, consisting of merging VT events, attributed to the excitation of fluids at relatively shallow levels, was observed one day prior to the onset of the explosive phase[14,15]. This observation suggested possible pressure oscillations within the ~6 km deep magma reservoir as the trigger of highly periodic tremor events. This strongly implied the imminent passage of gas-rich material from depth into the shallower edifice, consistent with the first detection of $SO_2$ flux on 8 April (Fig. 6). Then on 9 April, gas rich magma reached the surface and conditions for explosive fragmentation were realised, which correlated with the observation of syn-eruptive deflation (Fig. 2).

The working conceptual model provided a robust framework against which these rapidly emerging data could be interpreted and understood. This working model illustrates the need to interpret scientific data in real-time to inform rapid emergency decision-making and the difference between theoretical models and critical interpretations that trigger real-world, life-preserving decisions. The conceptual models, synthesised from quantitative data, were necessary for decision-making and formed a framework to create actionable evidence for responding to an acceleration in activity. Nonetheless, the conceptual model was also strongly informed by quantitative outputs from generic models for different aspects of volcanic behaviour and the input of boundary conditions obtained from new knowledge of the magma composition and observations of dynamic behaviour. Further scientific analysis with longer-term research programmes and quantitative modelling will test and improve these models. An important dimension of fully quantitative models is the recognition of generalizable insights relevant to other settings worldwide where rapid transitions in activity occur, that can be derived from empirical observations in real time.

**The role of uncertainty and impact of monitoring in a resource-constrained setting.** Interpretation of the monitoring data and development of a preliminary conceptual model were associated with large uncertainties when anticipating eruptive behaviour throughout the unrest episode. These uncertainties created both temporal (when) and spatial (how big) challenges. Specific uncertainties included: (i) interpreting the extent that seismic unrest patterns were similar to historical background seismic activity at La Soufrière volcano; particularly long episodes of unrest preceding >VEI4 explosions; (ii) during steady-state dome growth, distinguishing monitoring signals indicative of a potential acceleration of activity from normal behavioural fluctuations, in the absence of any significant measured deformation; and (iii) during the explosive phase anticipating the likely duration and peak intensity of explosions, given the range in size and documented intensity of the previous eruptions[8]. This contributed to uncertainties in interpreting signals that might represent the onset of an explosive phase and reduced timescales over which accelerations or decelerations in the intensity of activity could be confidently attributed to changing behaviour. Coping with uncertainties framed our conceptual model and attendant different scenarios. Our combined expert view of

likelihoods captured via expert elicitation, fed into decision focussed advice (e.g., VALS for La Soufrière). This approach avoided interpretations dependent on single outcomes[19], which inadvertently minimised aleatoric uncertainties or ignored ambiguities in datasets. The St. Vincent case demonstrates the benefits of the structured expert elicitation methodology to capture uncertainty.

An important factor in generating epistemic uncertainty was the relatively sparse monitoring network at the onset of the eruptive episode, which was a direct consequence of financial constraints on the monitoring operations. Limitations in the density of the network challenged our ability to definitively say whether the onset of the effusive eruption would have been instrumentally detected. However, as the monitoring network strengthened, observed signals were interpreted against improvements in data volume and accuracy. For example, additional GPS stations installed during the eruption greatly improved the sensitivity of the network. A sensitivity study[40], demonstrated that the network had no significant azimuthal gaps, but suffered from a lack of near-field stations to capture shallow deformation sources. In addition, interpretation of the low amplitude banded tremor detected on 8 April, reinforced by the observation of a detectable $SO_2$ gas flux later that day, was the most salient information to feed into changed views on explosion likelihood. Similarly, detailed seismic analysis and near real-time satellite and deformation measurements contributed to the anticipation of waning explosive activity during the acute phase (9–11 April).

The relatively late detection of effusive eruption onset and the importance of new data during the eruptive transition, clearly demonstrate that well-resourced multi-parametric networks are of high value. The reality in settings like St. Vincent is often different and network strengthening took place during the eruption, creating important safety concerns. SRC used a fieldwork life-safety risk assessment[41] with an estimated hourly risk of fatality exceeding $10^{-4}$ during the initial fieldwork period (see Supplementary Information). This procedure gave strong justification for the use of a helicopter and provided an opportunity for monitoring scientists to express any concerns and contribute recommendations on the best field practices.

**The value of collaborative preparedness, awareness and communication.** Identifying local needs and obtaining evidence of the efficacy and impact of the SRC's risk communication in the vulnerable communities[42] was a persistent challenge largely due to the Education and Outreach (E&O) team's remote operations. Harmonisation of messaging was essential[43]. Close collaborations with NEMO strengthened communication efficacy and reinforced local capacity for effective communication. In turn, this provided SRC with insight into appropriate content for its communication products. These types of relations take time and resources. The groundwork was essential to the success of managing the crisis in a rapidly changing volcanic situation with a requirement for the implementation of advice into action.

Wide acceptance of the risk information was indicated by the authorities acting decisively on advice provided by SRC, resulting in increased alert levels and the issuance of evacuation orders 24 h ahead of the first explosion. Furthermore, the public understood the increased volcanic activity and complied with evacuation orders.

**Integrated approach: value in anticipating eruptive transitions.** Effective crisis science and consequent volcanic risk reduction is a partnership between scientists, response agencies and the affected communities[44,45], It begins with the robust gathering and interpretation of scientific data, before, during and after a crisis. Our

analysis provides an excellent case study of the principles outlined in recent synoptic analyses[6,33].

However, important challenges arise in the acute crisis phase where decision-making timescale appropriate to the lifetime of the eruption (typically weeks to months) contract into minutes and hours with the growing prospect of a change in behaviour. As a transition threatens, uncertainty rises and demands dynamic interpretation of emergent datasets. Thus, our analysis here particularly reflects on the important drivers of risk in this moment.

An important dimension was the capacity to interpret data against a flexible conceptual model that expresses and formalises uncertainty. Further, the understanding from previous research[7–9,13,22] and agency-to-agency interactions that framed social context and societal constraints were important. Monitoring agencies need to be responsible for interpreting datasets and anticipating changes on societally relevant timescales. This responsibility also underpinned our communication strategies and timescales. The long-term relationship we described here increased the chances that advice given during an eruptive transition was more readily translated into actions by local emergency managers, and in turn, the populace at risk.

Research that accounts for the realities of managing crises could further improve effective decision-making. In volcanology, counterfactual analysis is a powerful way to understand what might have transpired[46]. A counterfactual analysis to include the range of possible scenarios and outcomes using the 'real time' evolving knowledge gathered, would assess whether the decision-making strategy here was robust to all eruptive outcomes. For example, considering situations where explosive activity happened at an earlier stage or explosions that generated larger pyroclastic density currents. Similarly, a focus on emerging petrological techniques that allow rapid forensic examination of timescales of disruption, degassing and ingress prior to other eruptive episodes would have significantly helped with the interpretation of changing monitored signals at the acute crisis point.

Finally, it is important to acknowledge the implicit risk to monitoring scientists during the intra-eruptive network strengthening. Our analysis demonstrates the value of the strengthened network, as well as remotely observed data, to data interpretation despite the risk in this particular case. Research that improves understanding of the effectiveness of monitoring networks would help identify strategies that best minimise risk, while maximising data benefit.

Operating in a resource-constrained setting influenced scientific response and emergency management. The steady global growth of disaster risk, volcanic or otherwise, compels disaster response agencies to fortify disaster preparedness capabilities and to ensure that institutional capacities are in place to optimize effective planning, response, and mitigation. Our assessment of the 2020–2021 La Soufrière eruption demonstrates the critical controls, produced over longer timescales, of an effective response during an acute crisis at the moment of eruptive transition.

Confidence in our conceptual models of reactivation via a gas-rich magma at depth was improved through the strengthening of the seismic network, real-time deformation and dome monitoring, changes in gas composition and petrological sampling. Nonetheless, as monitored signals shifted and the likelihood of transition increased, longer-term preparedness measures allowed us to disseminate rapidly changing information effectively on short timescales and contextualise our advice on timescales appropriate for actions to prevent loss of life, while minimising impacts on livelihoods.

## Methods

**Seismic monitoring**. The seismicity routinely used to assess the status of La Soufrière volcano derives from an eight-station network on and around the

volcano. Daily event counts are used to recognise changes within the system. The rapid densification of the network in early January 2021 (Fig. 1) facilitated the recording of micro-seismic signals generated by the dome emplacement process, as well as the detection and location of VT earthquakes. The location inversion was performed using a generic volcanic velocity structure[47], although this velocity model is not a result of 1D tomography, it provides consistent and clustered results when no shallow velocity structures are identified. The size of the remaining volcano earthquake types that could not be located was assessed by tracking the number and distance of stations recording those events along with the duration of the events as recorded by the crater rim station, SSVA and then by SVV. The recorded events were identified and processed by a team of seismology technicians at SRC and cross-checked with the seismologist on duty at the Belmont Observatory. An automatic event detection system was introduced after several weeks to support the analysis. Routine RSAM and spectral analysis calculations were also used in assessing the status of the system.

**Ground deformation monitoring**. GPS data were collected using Trimble NetRS and NetR9, and Septentrio PolaRX5 dual-frequency receivers and processed using GAMIT/GLOBK software (version 10.71)[48]. EDM were captured from six base locations (Fig. 1) in collaboration with the Lands and Surveys Department, SVG using a Leica Flexline TS06 total station. Radar imagery was acquired from Sentinel-1 satellites of the European Space Agency (ESA) and the ALOS-2 satellite of the Japan Aerospace Exploration Agency (JAXA). The ALOS-2 images were originally made available under an ALOS-2 6th Research call project[49] and were then also made available through an emergency collaboration with NASA. Formal requests were made to ESA and JAXA for additional collections, which were subsequently granted. Sentinel-1 repeat times were increased from 12–18 days to 6 days and ALOS-2 repeat times were increased from approximately annual to every 14 days. ALOS-2 data were processed using the GAMMA software[50] and topographic corrections were made using the 30 m ASTER GDEM. Sentinel-1 data were processed with the NSBAS processing chain[51,52] which relies on the legacy software ROI_PAC[53]. Topographic corrections were made using the 1 Arc-second SRTM DEM, and atmospheric corrections were performed using ECMWF's ERA-5 meteorological reanalysis[54].

**Gas and geochemical monitoring**. A portable Multi-component Gas Analysing System (MultiGAS) instrument composed of an infrared spectrometer and electrochemical sensors (plus air temperature, atmospheric pressure, and relative humidity sensors) allowed detection of the in-plume concentrations (ppm) of $H_2O$, $CO_2$, $SO_2$ and $H_2S$[55]. The instrument consists of a Gascard IR spectrometer for $CO_2$ determination (calibration range: 0–3000 ppmv; accuracy: ±2%; resolution: 0.8 ppmv) and of City Technology electrochemical sensors for $SO_2$ (sensor type 3ST/F; calibration range: 0–200 ppm, accuracy: ±2%, resolution: 0.1 ppmv), $H_2S$ (sensor type 2E; range: 0–100 ppm, accuracy: ±5%, resolution: 0.7 ppmv) and $H_2S$ (sensor type EZT3HYT; range: 0–200 ppm, accuracy: ±2%, resolution: 0.5 ppmv), all connected to a Campbell Scientific CR6 datalogger. The acquired data were post processed using the Ratiocalc software[56] with $CO_2/S_t$ ratios expressed in molar ratios,

Rock samples were collected directly from an active lobe of the dome on 16 January 2021, using a bucket. These were crushed and analysed from bulk composition using XRF. Subsequently, samples of scoria (erupted 9 April) and blocks from PDCs (emplaced 13 April) were also analysed. The dome samples were thin sectioned by Jesús Montes Rueda at the University of Granada and by Ian Chaplin at Durham University, and carbon coated. Scanning Electron Microscopy (SEM) imaging with Energy Dispersive Spectroscopy (EDS) analyses were conducted at the University of East Anglia (Zeiss Gemini 300 field emission SEM with Oxford Instruments Ultim Max 170 EDS). Imaging and analysis were conducted at 10 kV (UEA) with a working distance of 8.5 mm.

**Dome volume monitoring**. Growth of the new lava dome was monitored primarily through the application of photogrammetry, using images acquired from the summit crater rim or aerial images from observation flights using fixed-wing aircraft, helicopters or consumer grade unmanned aerial vehicles (UAVs). Images were processed using either ImageJ or the photogrammetry software package AgiSoft Metashape, the later used to generate 3D models of the lava dome from which volume and extrusion rates were determined. Due to the lack of a quality pre-eruption DEM, it was assumed that the dome had a purely flat base and exhibited either a pure hemispherical or half ellipsoidal shape. In reality, where the new lava dome reached the 1979 lava dome and the inner slopes of the Summit Crater wall, the dome had a slightly trapezoidal cross-section. Consequently, the volume data presented in Fig. 2 is overestimated by as much as 20%. The photogrammetry surveys were conducted at intervals of up to 34 days due to access and safety concerns (they were conducted from locations along the rim of the Summit Crater). Between surveys, radar and multispectral imagery from the Sentinel-1 and -2 satellite constellations and from Planet.com were used to track the extent of the footprint of the new lava dome.

**Hazard and risk evaluation**. Two complementary activities were undertaken to quantify anticipated risk from the La Soufrière and provide an evidence base for

internal decision-making during the eruption. The first, a fieldwork life-safety risk assessment provided estimates of the chance of fatality from an unheralded explosive event, which was a concern during the initial stages of the eruption when network strengthening fieldwork had to be conducted. The second, a formal approach to eliciting expert judgement, provided quantitative estimates of the likelihood for anticipated eruption scenarios that could inform both the fieldwork life-safety risk assessment and the provision of advice for emergency response and public safety throughout the eruptive sequence. This was undertaken on a regular basis to quantitatively assess the evolution of volcanic activity and possible future scenarios.

The fieldwork life-safety risk assessment was conducted following the VoLREst methodology[37]. The two-step procedure involved: (1) establishing the volcano-specific parameters, e.g., vent location, sites of interest, hazards of concern, eruption size categories, probability of exposure, probability of fatality and threshold of acceptable risk and (2) estimating eruption probabilities. Ideally the first step is undertaken in advance of any activity, in this instance parameters were identified after the extrusive eruption commenced, including elicited probabilities of exposure and fatality. Probabilities for step two were taken from the expert-elicitation for anticipated eruption scenarios. These values are combined in VoLREst to calculate hourly risk of fatality with increasing distance from the volcano (see Supplementary Fig. 4).

A structured elicitation process was initiated on 7 January 2021 to provide a framework for estimating quantitative probabilities of different eruption scenarios, particularly the likelihood of escalated eruptive (explosive) activity. Given that the eruption had already commenced, with extrusion at the surface in the form of a dome, three possible outcomes for the next stage of volcanic activity were considered: (i) effusive activity continues; (ii) eruption ends; and (iii) escalation to explosive activity. These scenarios formed the core of the questions during the elicitation, with probabilities elicited on a biweekly basis, with flexibility in adjusting the timing and content to address changing volcanic conditions, additional monitoring data, and/or questions that arose (both internally and externally) regarding possible scenarios.

The elicitation process included a briefing that was held approximately every two weeks to provide updates on the status of the volcano and monitoring operations. Previous elicitation results were discussed in depth during each meeting, together with a review of the scientific working model of the volcano and its ongoing eruptive state, and finally any possible changes to the elicitation questions. The group was then elicited immediately following the meeting, such that the estimated probabilities were based on consistent information available to all participants. Participants were asked to provide estimates of the median likelihood of a given event in a set time period, as well as estimates of 5 and 95% quantiles, to provide uncertainty ranges on their values. The Excalibur software package[19], which implements Cooke's "Classical Model"[18], was used to undertake the calculations. Estimates for the following one-week period and one-month period were elicited.

### Early warning and preparedness.

The Volcano-Ready Communities in St. Vincent Project (VRCP) was a grant funded community-based capacity-building programme that aimed to reduce vulnerability to the multi-hazard environment of the La Soufrière volcano across twelve (12) communities in St. Vincent. It was executed during the period April 2018 to November 2021 through collaborations involving the SRC, NEMO, the Red Cross Society and the Community Development Division of St. Vincent and the Grenadines.

Project activities were designed to enhance community early warning procedures; increase adaptive capacities; strengthen awareness; and enhance response capacities to enable community residents to effectively plan, prepare for and respond to the impacts of volcanic and other hazards. Activities included the production of a variety of print and digital public awareness and education materials (posters and brochures, film, photographs and public exhibits) disseminated through a series of multi hazard, gender-sensitive community sessions that facilitated public engagement. Community awareness and education materials included documentaries on best practices and lessons learnt from the 1979 eruption, were also captured through story telling in film, animation and photography. A total of four one-week educational sessions were conducted between the months of April 2018 and October 2019, involving both secondary school students and volunteers (32–45 participants per session). Other public awareness education activities included a crisis management scenario workshop, attended by 120 Fourth Form geography students, where students also participated in practical experiments that demonstrated the science of volcanic eruptions. In addition, a group of 80 Fourth Form students took part in a guided field visit to the volcano's summit, where they were introduced to SRC's volcano monitoring mechanism for La Soufrière volcano and its evolution since the 1979 eruption.

In addition, training and Workshops were conducted with community volunteers to develop community level volcano emergency plans for the citizens in the high-risk Red Zone of the Soufrière Volcano. Two workshops: (1) Initial Damage Assessments (IDA) and (2) Vulnerability and Capacity Assessments (VCA), were held with fifteen persons from seven communities located in the Red Zone participating in both workshops. The group consisted of four males and eleven females, with eight participants comprising of young adults (15–24 years). This facilitated incorporation of community hazard maps and databases identifying and mapping vulnerable persons, human and transportation resources for each community to be integrated into the national response mechanisms.

Community Emergency Response Teams (CERTS) certification training was also conducted for each community, with a total of 72 community volunteers being trained under this program. Participants were instructed on disaster preparedness for volcanic and other hazards that may impact their community and trained in basic disaster response skills, such as fire safety, light search and rescue, team organisation, and disaster medical operations. In addition, participants also received information on: Introduction to Disaster Management, Mass Care, Damage Assessment and Shelters and Shelter Operations. In addition, CERT teams were provided with personal and community emergency response tools and equipment upon completion of the training. Stakeholders (government, civil society, private sector) were engaged to assist four communities with the development and identification of resources for the implementation of the community response plans and provide support to test the National Volcanic Emergency Response Plan during Tradewinds 2019[38].

### Risk communication.

One of the key functions of the SRC is to provide information and scientific advice to governments, as well as to a large body of disaster management stakeholders and the general public. This was achieved through regular updates on volcanic activity as well as monitoring plans and techniques. At the onset of the eruption, the main objectives identified to guide the risk communication strategy were (i) to reinforce capacity of local authorities to communicate effectively; (ii) to promote public recognition of primary sources of information and (iii) to facilitate public understanding of science related to ongoing volcano monitoring techniques, volcanic activity, potential hazards and hazard mitigation measures.

The Education & Outreach (E&O) section of the SRC set out to reinforce NEMO's capacity by supporting the implementation of its communications plan. This was executed through regular consultations between the two agencies to share communication expertise and collaborative hosting of key public education activities. The SRC developed communication products to address specific areas of need.

As part of intensified efforts to reinforce public recognition of primary sources of information, SRC spokespersons were identified for the eruption, visibility and responsiveness on social and traditional media were increased and published communication products were branded with SRC logo. A standardized statement identifying SRC and NEMO as official information sources were integrated as a consistent message across products, including interviews.

Where possible, communication products contained jargon-free language and alternatively, simple explanations for technical terms were provided where the scientific language was unavoidable. A customised communications approach to address different learning styles was adopted and information was disseminated through multimedia to targeted audiences. Eruption-related questions trending on SRC social media platforms provided the basis for these user-informed products.

## Data availability

The raw datasets that informed the analyses presented in this study are available from the corresponding author on request. Sentinel products were freely downloaded from the Copernicus Open Access Hub (https://scihub.copernicus.eu/).

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

## Acknowledgements

For their support to the monitoring of the eruption of La Soufrière, the staff of the SRC and MVO are thanked. Roderick Stewart is thanked for his service as a Scientific Team Lead in St. Vincent. We thank the Director and staff of NEMO, the SMU, Lands and Surveys Department of St. Vincent and volunteers assisting with field work for campaign EDM and GPS occupations. Willy Aspinall is thanked for sharing his insights and expertise throughout the unrest and eruption episode. We thank the ESA and JAXA for tasking systematic Sentinel-1 and ALOS-2 acquisitions over La Soufrière. We thank the Capella Space and ICEYE companies for providing high-resolution SAR images of the volcano during the explosive phase. ALOS-2 SAR data were made available by the Japanese Aerospace Exploration Agency (JAXA) for the sixth RA proposal (PI no. 3153). Collaborating agencies such as NASA, IPGP, LOA, AERIS/ICARE, CSIC, USGS VDAP, USAID, CIMH, KNMI, CEOS Volcano Demonstrator all provided support for remote sensing data analysis. We thank the MOUNTS platform for their automated analyses of Sentinel data for La Soufrière, SVG. Fieldwork for JB and PC and petrological analysis was supported by NERC Urgency Grant NE/W000725/1 and Royal Society Apex Award APX\R1\180094. We thank David Pyle, Gregor Weber and Jon Blundy for assisting with geochemical analysis of dome samples. Funding for the VRCP project was granted to RR (Grant No. GA 43/STV) through CDB's Community Disaster Risk Reduction Fund (CDRRF) and is supported by the Government of Canada and the European Union. We also thank the Executive Director and staff of CDEMA for logistical support and resource mobilisation efforts.

## Author contributions

E.P.J. took the lead in writing the manuscript and oversaw the overall management of the UWI SRC during the volcanic crisis. M.C.-.H., T.C., A.S., R.R., J.B., L.L. and P.C. conducted fieldwork, collected data, performed analysis and contributed to the interpretation of results. V.L.M. collected data, performed analysis and contributed to the interpretation of the results. K.P., G.R., J.L.L., R.C-.A., M.J. and I.P. analysed data and contributed to the interpretation of the results. S.E., O.G. and A.J. contributed to the design and implementation of the outreach research. R.G., I.H. and M-.J.J. analysed data and contributed to the interpretation of the results. J.B., P.C. and B.V.D. analysed the dome rock and eruptive products and contributed to the interpretation of the results. R.S.J.S. contributed to the development of a working conceptual model of the volcano. All authors provided feedback and helped shape the research, analysis and manuscript.

## Competing interests

The authors declare no competing interests.
