## [Peer Review File · Nature Communications]

REVIEWER COMMENTS

Reviewer #2 (Remarks to the Author):

According to its title, this manuscript is an account of the scientific response during the 2020-21 eruption at St. Vincent island. The account describes the implemented monitoring system, the observations recorded, and the communication with stakeholders (public officials and decision makers, the media and the public). The volcano was poorly monitored, but when the unrest started, the monitoring system was improved. The scientific response was such as to lead to quick evacuation of the areas subject to risk, and no losses were reported.

The account tells a story which is not dissimilar from others characterizing active volcanoes in settings with limited resources. The scientific content is rather low and far from the standards characterizing papers published in journals with high impact. The reflection on managing volcanic risk in a not highly developed setup does not bring much, as similar contents are found in other such accounts (e.g., those relating to a number of recent eruptions from Indonesia published in specialized journals). Sincerely, I do not think this account is of any interest sufficient to justify a publication in Nature Communications.

Below I report more specific comments.

In general, the risk response organization does not emerge clearly, as it is not fully clear who is responsible for what. For example, at line 536 it is said that "Protocols to alert the whole population [...] were developed with the target communities etc.". That does not clarify who developed the protocols for alerting the population, what was the related role of scientists, and finally if scientists are expected to alert themselves or have a role in alerting the population. The adopted VALS (reported in the Supplementary Information) does not help, as it presents direct relationships between scientific evaluations and actions on the population (once you have the former, the latter is also determined); contributing to depict a system where scientists may effectively contribute to make decisions for the population. That may be the case, but it would be worth saying it clearly – or if it is not, then avoid any source of possible misunderstanding.

Line 90: UWI is introduced without definition (I guess it is University of West Indies).

Line 121: Fig. 1 is quoted in relation to an "increase in seismicity in November 2020"; however, the figure does not include that period.

Line 128: what is the meaning of "similar" in this context? The frequency of the second group of events is up to one order of magnitude smaller, which does not seem to justify the conclusion that they are "similar". On the other hand, there are no other characteristics reported, so the statement remains unclear.

Line 130. Fig. 3B is quoted in relation to "Hypocenters" which "delineated a NW-SE structure". However, the figure does not show hypocenters, only epicenters.

Fig. 4A. The claimed "banded tremor" cannot be clearly appreciated under the thick lines drawn to evidence it. That is reported as the suggestion for "an imminent explosive phase onset"; this seems to deserve a clearer picture and more thorough discussion.

Line 143. The claimed "same stable frequency content" from the spectrograms is unclear from the figure, and not supported by any quantitative analysis.

Line 145. There is a claimed "exponential decay" in Fig. 4, but no explanation on how such a

conclusion is reached.

Line 154. The note on the shallow (about 500 m) character of the magma intrusion relates to the recorded amplitude, which looks weird without analysis of the geometry of the deformation. Please explain. Related to this point: at line 161 another phase of deformation is said to be associated to a 6-7 km deep source. Is the geometry of the deformation well constrained, and if so, why not showing it? The following line 162 reports an estimate of the "overall erupted volume" (over about 10 days) which appears to compare with the deformation "in the first 24 hours..."; what is the relationship between the two estimates, which refer to time spans differing by one order of magnitude?

Lines 167 and followings (beginning of section 1.4), and Fig. 2. Geochemical ratios are reported without reference units. Are these mass, molar, volume or other ratios?

Line 179. The inferred viscosity is surprisingly high, comparable with that of dry rhyolite and definitely not typical for basaltic andesites. The reported reference for typical viscosities at La Soufriere is that of Cole et al. 2019, but those authors do not even mention viscosity in their paper. Please explain.

Section 2 on Crisis Response. I have outlined above my major concern relating to unclear definition of roles and responsibilities. I add that this section includes a higher proportion of sentences whose English is poorly effective contributing to generate confusion.

Line 223. The reported source of uncertainty ("imperfect understanding") is only the epistemic component of uncertainty. There is another component which is not mentioned and can be equally or even more relevant. This is the aleatoric component of uncertainty, generated by the complex behavior of highly non-linear volcanic systems. Overlooking the aleatoric component of uncertainty may be dangerous when interpreting complex sequences of data, as it may lead to over-interpretations and poorly justified conclusions.

Line 232. Reference to "central value" is ambiguous, as there are many measures of the central tendency of a distribution. The median is cited two lines above, so I guess that's the referred quantity? Please clarify.

Line 234. Here it is said that the elicitation process led to a strong increase (from 20 to 60%) in the (median? Please clarify) probability of explosive activity in the morning of 8 April. What were the observations that led to such a dramatic change in the evaluations by the elicited experts? Fig. 2 seems to show that no major changes occurred on April 8 (VT seismicity had increased in the previous days and does not seem to show anything major on 8 April). Low amplitude tremor is reported, is it the major cause of the jump in probability?

Line 234: SVG (Saint Vincent and the Grenadine, I guess) is not previously introduced.

Line 243 (and others similar): put a space between numbers and their dimensions.

Discussion section. This is all highly speculative, the kind of conceptual modeling any volcanologist extracts from sequences of data and observations, in this case with relatively limited data. Fine, but reinforcing the clear impression that Nature Communications is not the appropriate journal for this contribution.

Reviewer #3 (Remarks to the Author):

please note - I will also upload a file of these comments which will be easier to read.

Lindsay review of "The 2020 – 2021 eruption of La Soufrière volcano, St. Vincent: Monitoring and Scientific Response" by Joseph et al.

Overview

This paper presents an overview of the monitoring data during the 2020 – 2021 eruption of La Soufrière volcano in St. Vincent, and how the data, combined with prior knowledge and experience of the volcano, were used to inform decision-making, crisis communication and assessment of risk. This is a very interesting manuscript that provides new insights into a very recent eruption, and that highlights some of the challenges of responding to volcanic crises, especially in resource-constrained contexts. I think it will be of interest to a wide audience as it provides a comprehensive summary of the eruption, the monitoring data that were obtained, and the actions that were taken – including the development of an evolving conceptual model. Overall it is a well-written paper and my comments are generally quite minor. I understand that papers published by Nature Communications “represent important advances of significance to specialists within each field”. Although this manuscript doesn’t present advances as such, I am of the opinion it is extremely important to document such eruptions, eruption responses, and the link between data streams and evolving conceptual models. If required for publication in Nature Communications I am sure the manuscript could be tweaked so that its focus is more on the novel aspects.

General comments and suggestions:

I found it unusual that the hazard zones in the hazard map were labelled in the key as red, orange, yellow and green, instead of an explanation of what the zones mean. That’s the equivalent of a key to a dotted area on a map just saying “dots” in the legend. Is this really the map that was used? It may be that the population on St Vincent are so familiar with the hazard zones that it is sufficient to just show the colours with no explanation – but I think for an international audience the original descriptions of the zones are required somewhere.

I wonder if it would be worthwhile providing the actual results of your life-safety risk assessment and how these evolved during the crisis—and an explanation for how this affected decisions made to undertake field work? This would be of interest to other observatories and agencies I am sure. Also – in the risk assessment section on page 23 it seems you really only describe in detail the expert elicitation related to the evolving eruptive processes (ie the second approach) – which is not a risk assessment in itself. Maybe make this section clearer by providing a bit more information about how the results of the expert elicitation fed into any actual risk assessments, including for example how (if at all) the expert elicitation that you describe here feeds into the life-safety risk assessment.

Specific comments:

Page 1 line 20 – “in writing the manuscript and was in charge.....”

Page 1 contributions section – be consistent about how you present peoples’ initials. Some have full stops some don’t

Abstract – line 42 – add 2021 after April.

Abstract – line 43, delete comma after forecasts. Change By to “In contrast”

Abstract – line 45, add hyphen – resource-constrained

Page 3 lines 50 and 51 – I got very confused with the two ones, superscripts. To avoid confusion, maybe remove the footnote and just define crisis science in the text.

Page 3 – definition in the foot note – this was confusing because the text said crisis science yet the footnote explained “science during crisis”. I would stick to the same wording if you are keeping the footnote – but suggest you get rid of it and explain in the text: “Crisis science is defined here as conducting.....”. Also – in the footnote you refer to this “report” – rather than “here” or this “paper”.

Page 3 – line 53 suggest a change to “.....and 300-600m deep4”

Page 3 line 61 – space after 1971.

Page 3 line 66 – remove comma after 2020

Page 4 line 82 – suggest deleting “also” before underpin.

Page 4 line 83 – well-documented (add hyphen). Also - I suggest saying “We also reflect on....”

Page 4 line 91 – should anglophone be with a small a?

Page 5 – lines 110 and 113 – I suggest adding 2021 after each mention of January

Page 6 lines 119-120. This first sentence is confusing. The seismicity associated with inter-volcanic episodes at La Soufrière is generally sparse, but interspersed by irregular, short-lived swarms. Is the sparse seismicity (whatever that looks like) really interspersed with irregular short lived swarms, or do the short lived swarms actually represent the sparse

seismicity ? Maybe rephrase.

Page 6 line 121 – suggest adding a 2020 after December.

Page 6 line 129 – There is no Fig 2A – I believe this should refer to 3A?

Page 6 line 129 – add 2021 after March

Page 6 lines line 130 - Figure 3 doesn't show hypocentres – rather epicentres. Change to epicentres

Page 6 line line 30 delete comma from after "swarm"

Page 6 line 134 – add 2021 after April

Page 7 line 143 – add "the" in "hours of the explosion"

Page 7 line 145 should the reference here be to 4B?

Page 7 line 146 – there is no Fig 4C – double check what figure you are referring to

Page 7 line 150 and 151 – can these two sentences be combined into one? Also – delete comma after (Fig. 1)

Page 7 line 155 – this sentence is confusing. "Subsequently no unambiguous deformation signal was detected from the SAR platforms"- that means that all subsequent deformation signals were ambiguous. Or do you mean – no deformation at all was observed?

Page 7. Line 157. It is impossible to see 9 April on the figure. Add ticks that can be seen, and maybe label 9 April?

Page 7 line 162 – what do you mean by analysis of dome removal? Should that actually read dome volume?

Page 8 line 167 – add 2021 after January, and add comma after instruments

Page 8 line 170 whange were to was. SO2 is singular

Page 8 line 173 traverse of..... what? A crater lake? In the sea off the coast under the plume?

Page Page 8 line 180 – delete brackets, this makes the sentence easier to understand.

Page 9 – line 200 – add 2021 after March.

Page 10 – line 214 delete comma after Fig. 1)

Page 10 line 228 – delete comma after March 2021

Page 11 line 232 – what do you mean by "central value"?

Page 11 line 243 – Maybe add a sentence about what happened between 12 April and 22 April? At present you jump straight to the end of the eruption without saying what happened in between.

Page 13 line 291 – Consider changing "construed" to a better word. Do you mean presented? Otherwise you are effectively saying "interpretations need to be interpreted in the context....."

Page 13 lines 293-295 This sentence is a bit clumsy. "Aims" don't require rapid response – "achieving aims" might.... The second part of the sentence is also unclear. "Rapid response" of monitoring data doesn't make sense, for example. Maybe – "In our experience, achieving these aims requires rapid collection of monitoring data that are then analysed, interpreted and communicated in light of the experience accrued from past activity at the volcano as well as long-term engagement with stakeholders and communities on the ground. However, you have to make sure that this isn't just a repeat of what is in the previous sentence, at the moment the way you have written it reads as saying the same thing in different words.

Page 14 page 312 – It is unclear to me why this process (injection of gas rich magma) would lead to low rates of seismicity. Maybe spell that out. Also – in line 310 – what is the gas rich magma being injected into? Do you mean – injected into a shallow reservoir?

Page 14 – line 313, I suggest changing "being largely focussed on" to "expanded to infer the presence of"

Page 14 line 314 – maybe "degassed magma that had remained within the conduit following the 1979 eruption"

Page 14 line 317 – add a "had" before reinforced

Page 15 line 323 – add commas after region and after crust.

Page 15 line 323 – "higher impact" eruption is vague. Do you mean larger eruption? More explosive activity? More voluminous activity? Longer duration activity?

Page 15 line 328 add 2021 after April, then changed halted to followed

Page 15 line 330 and 331. It sounds like you are talking about hypocentres but you say epicentres. Just check that you actually do mean epicentres (and maybe refer to the figure that shows the change in epicentral patterns if that is indeed what you are referring to)

Page 15 line 340 VT events (plural)

Page 15 line 344 – maybe remind us here what was happening with gas.

Page 16 line 346 – Refer to the figure after deflation
Page 16 line 359 – conceptual model rather than models?
Page 16 Line 362 – the signals themselves were not uncertain. Maybe change to unclear?
Inconclusive?
Page 16 line 362- 363 maybe change to “(iii) once explosions started the the extent to which they might continue was unknown, given the range in size and intensity of previous explosions”
Page 17 – line 372 this sentence doesn’t make sense. Maybe delete “understanding”?
Page 17 lines 373 onwards – maybe provide an actual example of a process and a link with monitoring data?
Page 17 line 386 – delete hyphen from field-practices
Page 17 line 388 “Than is acceptable in other..”
Page 18 line 401 – should the IAVCEI reference have a superscript? Maybe give the title and then a superscript to the full reference in the reference list
Page 18 line 409 – Perhaps the best indicators (plural – you present more than one) also change was to were
Page 18 line 410 – the way this is written could be interpreted to mean the authorities acted decisively by raising alert levels. Maybe tweak so it is clearer?
Page 19 – line 415 – change consequently to consequent
Page 19 line 418 – maybe say “the monitoring network” rather than just the network. Also perhaps change to: The rapid strengthening of the monitoring network, the continuous stream of communication between scientists and authorities, and the use of a common framework to anticipate changing behaviour are key factors. (also – do you mean common framework – or conceptual model?)
Page 19 – line 423 – perhaps start this sentence with “In the case of the 2020-2021 eruption of La Soufriere in St Vincent, we propose that.....”
Page 19 line 425 – maybe delete “and work”
Page 19 line 427 – all-important conceptual models? (otherwise you could be talking about computer or numerical process models)
Page 19 line 429 – maybe modify to something like “.....heightened activity is more readily translated into actions by local emergency managers, and in turn, the populace at risk” (I don’t think health and safety officials is the right phrase here)

Methods

Page 20 Line 441 – delete comma afte generated
Page 23 – as mentioned above, it would be interesting to see how the results of the life safety risk assessment evolved during the crisis. Also – in this section it seems you really only describe in detail the expert elicitation related to the evolving eruptive processes (ie the second approach) – which is not a risk assessment in itself. Maybe make this section clearer by providing a bit more information about the life-safety risk assessment – including for example how (if at all) the expert elicitation that you describe here feeds into it. And, how the results fed into any actual risk assessments.
Page 24 line 536 – delete comma after population
Page 24 line 538 and elsewhere – what do you mean by gender-sensitive materials? Maybe explain this somewhere?
Acknowledgements – line 586 delete “eruption” after soufriere. Also – field work for JB and PDC – unclear who PDC is given Paul Cole doesn’t have a middle name in the list of authors

Figures

Figure 1

– See comment about about the strange legend. What do the hazard zones actually show?
– The red triangles are hard to see against the red background. Maybe give them a black outline? (probably good for the green circles too)
– Maybe change the first sentence in the caption to: “Volcanic hazard map of LS showing the monitoring network” or – “Volcano monitoring network for La Soufriere, St. Vincent plotted on the volcanic hazard map”

Figure 2

- C/Stot – needs explaining in the caption. Also use the subscript for tot.
- Improve the x axis labels – how can a single tick represent a whole month? Are they supposed to indicate the first day of each month? If that is the case, then that should be the label. I suggest also including some lower-level ticks, and labelling / annotating more key

dates (for example start and end of explosive phase)

Figure 3

- You could probably be a bit more specific in B and C and say these are epicentral plots

Figure 4

- The minor ticks on the x axis are not visible – maybe make them bigger? Otherwise it is very hard to estimate what date falls where in the plots.

- There is no C in this figure yet you refer to it in the text.

Figure 6

- Fix the subscripts in the figure? And the caption

Figure 10

- I found the dots in this figure confusing. Maybe in the caption provide an example of “how to read” the table, by explaining what the dots mean in the context of one single entry?

Supplementary material – It would be good to have more detail in the captions about what we are actually looking at in these two figures. What was the first flyer used for? Who was the audience? In the second figure – who is in all the figures? The journal will likely require written permission from everyone in the photographs in order to publish them. What are we looking at exactly? Are we looking at screen shots of radio interviews? You tube videos? Please explain in the caption.

Reviewer #1 (Remarks to the Author):

Overall

This was an interesting paper on how an observatory translates science into actionable forecasts and hazard assessments, and how it communicates these with the public. This is a topic that deserves more attention, as it is a critical (perhaps the most critical) part of an eruption response. As the majority of attention in the literature is usually devoted to fundamental science, this paper helps fill in an important gap in the field. My main comment is that the article should spend more time on the broader implications of this eruption response, and how it might inform crisis response elsewhere – right now the scope of the paper is too limited. My other comments are all minor, and include reading through the manuscript to fix an assortment of awkward sentences.

Main comment

In my reading, the paper details the 2020-2021 La Soufriere response and the successful aspects of it, but stops short of explicitly presenting broader implications from the experience and how these insights can be applied elsewhere. I'm assuming for this journal, in particular, there should be a significant portion of the paper devoted to what this response teaches us in a broader sense. The introduction does a good job of setting up the motivation, but the discussion and conclusion seem to fall short on relating this study to the outside world. These sections invoke some general statements about volcano monitoring, but I think the reader will want more explicit or detailed links between this study and other volcanoes. I was expecting some discussion of how this response might relate to eruption crises at other volcanoes (right now there's very little mention of other volcanoes or eruptive crises), and how the lessons learned here might be applied in other scenarios.

Specific comments

- Line 234: Here it is stated that the elicited probabilities rose dramatically, and it would be helpful to note briefly the reason – the appearance of banded tremor, correct?
- ~~Line 237: It would be helpful here to note again the dates when the explosions occurred.~~
- Line 262: The comment about harmonizing the messaging between the observatory and emergency responders is an important one. It brings to mind recent activity we were involved in here in Hawai`i, and this paper might be relevant in how communication can be done in the midst of an eruption crisis:

<https://pubs.geoscienceworld.org/books/book/2116/chapter/115281994/Communication-strategy->

of-the-U-S-Geological

- Line 335: It's interesting that the flushing of early magma followed by fresher magma, with the associated increase in eruption vigor, is observed in this eruption and also in other environments like Kilauea. The Gansecki et al. paper below also describes how this shift in composition implied a change in eruptive vigor, and how this concept was communicated

with emergency responders as the events unfolded:

<https://www.science.org/lookup/doi/10.1126/science.aaz0147>

- Line 345: This is an example of an awkward sentence, and there are multiple examples of this throughout the paper. One way to avoid sentences like this is to read through the paper out loud and find where the reading gets tripped up. I know I was getting tripped up a few places.
- Line 360: The wording here is unclear, especially for item (i)
- Line 580: This is vague, and I'm not sure exactly what is meant here.
- Figure 10 is an excellent communication summary. It's a busy graphic, but it contains the needed information.

THE UNIVERSITY OF THE WEST INDIES
SEISMIC RESEARCH CENTRE
ST. AUGUSTINE, TRINIDAD AND TOBAGO, WEST INDIES

24 February 2022.

We wish to thank the reviewers and the editor for their time and consideration of our manuscript. We have addressed the referees' comments as detailed below (referee comments shown in italics). All minor corrections have been accepted, text (see 'track changes' version of the resubmitted manuscript) and figures and tables have been modified to accommodate the suggestions of the reviewers. **The line numbers mentioned in the responses below refer to the changes marked version of manuscript.**

We are particularly grateful for the editorial steer around the contrasting reviews. The key focus here is on the 'crisis science' - which is what reviewers 1 and 3 have pointed out. This focus perhaps speaks more clearly to the ambition of Nature Communications than the tradition of scientific focus some readers might have for 'Nature'. We are grateful for the opportunity to help achieve that ambition. However, from reviewer comments, this needed to be clarified and developed in the responses. We have focused on strengthening relevance to other situations worldwide, particularly for volcanoes that experience transitions from effusive to explosive eruptions and the key findings of relevance to 'science into action'. The detailed comments on their reviews (and those of Reviewer 2) have helped with this. An important dimension here, which we now develop more fully in the discussion is pushing the frontier of this work in a resource-constrained setting.

We summarise here the contributions to this:

- (a) **in the introduction we have more clearly introduced the concept of crisis science, its relationship with operational volcanology and the value of the La Soufrière experience to the wider volcanic community.**

Reviewer #1:

Review of "The 2020-2021 eruption of La Soufriere volcano, St. Vincent: Monitoring and scientific response" by Joseph et al.

Review by Matt Patrick, USGS-HVO, mpatrick@usgs.gov

Overall

This was an interesting paper on how an observatory translates science into actionable forecasts and hazard assessments, and how it communicates these with the public. This is a topic that deserves more attention, as it is a critical (perhaps the most critical) part of an eruption response.

As the majority of attention in the literature is usually devoted to fundamental science, this paper helps fill in an important gap in the field. My main comment is that the article should spend more time on the broader implications of this eruption response, and how it might inform crisis response elsewhere – right now the scope of the paper is too limited. My other comments are all

minor, and include reading through the manuscript to fix an assortment of awkward sentences.

Main comment

- In my reading, the paper details the 2020-2021 La Soufriere response and the successful aspects of it, but stops short of explicitly presenting broader implications from the experience and how these insights can be applied elsewhere. I'm assuming for this journal, in particular, there should be a significant portion of the paper devoted to what this response teaches us in a broader sense. The introduction does a good job of setting up the motivation, but the discussion and conclusion seem to fall short on relating this study to the outside world. These sections invoke some general statements about volcano monitoring, but I think the reader will want more explicit or detailed links between this study and other volcanoes. I was expecting some discussion of how this response might relate to eruption crises at other volcanoes (right now there's very little mention of other volcanoes or eruptive crises), and how the lessons learned here might be applied in other scenarios.

We have attempted to address this in the revised manuscript by refocusing the paper, particularly in the Discussion section (lines 476 – 508), around responding to transitions from effusive to explosive eruptions, a challenge at many volcanoes worldwide and the institutions responsible for monitoring them. Global review of recent historical eruptions (Barclay et al. 2019), robustly demonstrates that it is in the eruptive transitions and the fog of uncertainty where decision-making mistakes (both institutional and individual) are made and there are complex contributors to that – of which one key lesson for those monitoring volcanoes is that forecasting needs to be actionable on timescales appropriate to those on which 'real life' decisions are made. This paper articulates that actionable forecast of eruptive behaviour are a critical challenge for monitoring scientists, particularly in resource-constrained settings, as well as taking into consideration the understanding of volcanic behaviour via observed signals and past knowledge of social systems (boiled down in times scales of decision-making and communication pathways in this context). This manuscript is a documentation of this happening in a robust evidence-driven way, with a strong secondary message of the additional challenges posed in a real-life setting. Some generalisable lessons that are now more explicitly discussed to show the broader implications of this eruption response, and how it might inform crisis response elsewhere, include:

- *How knowledge of volcanic history and monitored signals drive conceptual models, which can help to recognise eruptive transitions (lines 349 – 418).*
- *How the use of working conceptual models could be used to inform the scientific response to emergency management and provide advice to the authorities, as well as anticipate explosive transition or other significant changes in activity (lines 421 – 439).*
- *Actionable forecasts need to account for timescales on which forecasts can be actioned to save lives but also be considerate of minimizing impacts on livelihoods and constraints of local authorities (lines 350 – 355).*
- *Crisis decisions made during short timescales are very strongly enhanced by knowledge developed over longer timescale (lines 372 – 375).*

- Line 234: Here it is stated that the elicited probabilities rose dramatically, and it would be helpful to note briefly the reason – the appearance of banded tremor, correct? *Correct, included in the revisions (line 272).*

- Line 237: It would be helpful here to note again the dates when the explosions occurred. *Done (Line 277)*

- Line 262: The comment about harmonizing the messaging between the observatory and emergency responders is an important one. It brings to mind recent activity we were involved in here in Hawai'i, and this paper might be relevant in how communication can be done in the midst of an eruption crisis:

<https://pubs.geoscienceworld.org/books/book/2116/chapter/115281994/Communication-strategyof->

the-U-S-Geological

Comment and reference included (lines 304 – 307).

- Line 335: It's interesting that the flushing of early magma followed by fresher magma, with the associated increase in eruption vigor, is observed in this eruption and also in other environments like Kilauea

The Gansecki et al. paper below also describes how this shift in composition implied a change in eruptive vigor, and how this concept was communicated with emergency responders as the events unfolded:

<https://www.science.org/lookup/doi/10.1126/science.aaz0147>

(see lines 381 – 383 and following discussion lines 391 – 404 on changes in eruptive activity, as well as section 2.3 on communication strategies throughout the eruption).

- Line 345: This is an example of an awkward sentence, and there are multiple examples of this throughout the paper. One way to avoid sentences like this is to read through the paper out loud and find where the reading gets tripped up. I know I was getting tripped up a few places.

Paragraph deleted; idea now included in lines 402 - 404.

- Line 360: The wording here is unclear, especially for item (i)
See revised sentences (lines 424 – 431).

- Line 580: This is vague, and I'm not sure exactly what is meant here.
See revised sentence (lines 707 - 708).

- Figure 10 is an excellent communication summary. It's a busy graphic, but it contains the needed information.

Figure 10 has been revised.

Reviewer #2:

Reviewer #2 (Remarks to the Author):

According to its title, this manuscript is an account of the scientific response during the 2020-21 eruption at St. Vincent island. The account describes the implemented monitoring system, the observations recorded, and the communication with stakeholders (public officials and decision makers, the media and the public). The volcano was poorly monitored, but when the unrest started, the monitoring system was improved. The scientific response was such as to lead to quick evacuation of the areas subject to risk, and no losses were reported.

The account tells a story which is not dissimilar from others characterizing active volcanoes in settings with limited resources. The scientific content is rather low and far from the standards characterizing papers published in journals with high impact. The reflection on managing volcanic risk in a not highly developed setup does not bring much, as similar contents are found in other such accounts (e.g., those relating to a number of recent eruptions from Indonesia published in specialized journals). Sincerely, I do not think this account is of any interest sufficient to justify a publication in Nature Communications.

As is noted by several learned academies and other organisations (eg. National Academies of Sciences Eng. Medicine. "Volcanic Eruptions and Their Repose, Unrest, Precursors, and Timing. The National Academies Press (2017)", IAVCEI and recent report (Loughlin SC, Sparks RSJ, Sparks S, Brown SK, Jenkins SF, Vye-Brown C. Global volcanic hazards and risk. Cambridge University Press (2015)) to the Global Assessment of Risk (UNDRR GAR) an integral part of success in forecasting and mitigating the

impacts of volcanic hazards is an integrated interdisciplinary team. As described in Chapter 3. of the recent National Academies Report, there are very little documented success let alone analyses of success in this domain. This paper represents a step change by offering that initial insight and describing what, at the time, informed decision making. We have used the lens of crisis decision-making to do this. This means our work is not only of value in volcanic contexts, but can also be considered in other long duration, varied hazard contexts. We suspect these comments around just the physical science arose because this was insufficiently clear, so we have set out to address this here.

Below I report more specific comments.

In general, the risk response organization does not emerge clearly, as it is not fully clear who is responsible for what. For example, at line 536 it is said that “Protocols to alert the whole population [...] were developed with the target communities etc.”. That does not clarify who developed the protocols for alerting the population, what was the related role of scientists, and finally if scientists are expected to alert themselves or have a role in alerting the population.

The adopted VALS (reported in the Supplementary Information) does not help, as it presents direct relationships between scientific evaluations and actions on the population (once you have the former, the latter is also determined); contributing to depict a system where scientists may effectively contribute to make decisions for the population. That may be the case, but it would be worth saying it clearly – or if it is not, then avoid any source of possible misunderstanding.

Sentences revised to avoid possible misunderstanding (lines 633 - 635). The word protocol was changed to procedures (line 641) to better reflect the meaning of the sentence, as it was not meant to refer to the actual alerting protocols, but the actions taken by citizens in the Red Zone when an evacuated order/alert is given by the local authorities (NEMO) - lines 471 – 475.

Line 90: UWI is introduced without definition (I guess it is University of West Indies).
Correct. Change made in line 102.

Line 121: Fig. 1 is quoted in relation to an “increase in seismicity in November 2020”; however, the figure does not include that period.

Sentence revised (see lines 128 – 130) and reference to Figure 1 removed. .

Line 128: what is the meaning of “similar” in this context? The frequency of the second group of events is up to one order of magnitude smaller, which does not seem to justify the conclusion that they are “similar”. On the other hand, there are no other characteristics reported, so the statement remains unclear.
Corrected: our analysis shows that the events share the same properties of low frequency events reported in literature (see lines 133-135).

Line 130. Fig. 3B is quoted in relation to “Hypocenters” which “delineated a NW-SE structure”. However, the figure does not show hypocenters, only epicenters.
Paragraph revised (see lines 131 - 138).

Fig. 4A. The claimed “banded tremor” cannot be clearly appreciated under the thick lines drawn to evidence it. That is reported as the suggestion for “an imminent explosive phase onset”; this seems to deserve a clearer picture and more thorough discussion.

More information and references were included in the text where banded tremor is mentioned. We highlighted the fact that this tremor is related to shallow fluid excitations (gas and hydrothermal), which is a sign of a transition to explosive phase (due to the lack of gases involved during the effusive phase). Lines 139 – 142.

Line 143. The claimed “same stable frequency content” from the spectrograms is unclear from the figure, and not supported by any quantitative analysis.

A spectrogram is a recognized form of quantitative analysis and Figure 4 was revised to depict the characteristics referred to more clearly. Sentence revised (lines 151 – 153).

Line 145. There is a claimed “exponential decay” in Fig. 4, but no explanation on how such a conclusion is reached.

Figure 4 zoomed in and RSAM included, each explosion has an exponential decay, it is a direct observation.

Line 154. The note on the shallow (about 500 m) character of the magma intrusion relates to the recorded amplitude, which looks weird without analysis of the geometry of the deformation. Please explain.

Related to this point: at line 161 another phase of deformation is said to be associated to a 6-7 km deep source. Is the geometry of the deformation well constrained, and if so, why not showing it?

The near-field deformation at the start of the effusive phase, which was only observed with InSAR data (there were no GPS station near the crater at the time), was best modelled using an Okada dislocation source, which is detailed in Figure 5’s caption. Conversely, modeling of the explosive phase using cGPS data, was best characterised using a Mogi point source model. We modified the text to ensure the two time periods and their corresponding deformation source modelled are now clearer to the reader (lines 176 – 178). More detailed modeling to constrain the source geometry will be addressed in another scientific paper and so we opted not to show this here due to the broader scope of this paper.

The following line 162 reports an estimate of the “overall erupted volume” (over about 10 days) which appears to compare with the deformation “in the first 24 hours...”; what is the relationship between the two estimates, which refer to time spans differing by one order of magnitude?

Sentence comparing volume of erupted material removed over different time spans was removed (lines 178 – 180).

Lines 167 and followings (beginning of section 1.4), and Fig. 2. Geochemical ratios are reported without reference units. Are these mass, molar, volume or other ratios?

These are ratios of concentrations in parts per million (ppm) (lines 184 - 189). See methodology section, lines 558 – 568 for more information on the geochemical measurements using MultiGAS.

Line 179. The inferred viscosity is surprisingly high, comparable with that of dry rhyolite and definitely not typical for basaltic andesites. The reported reference for typical viscosities at La Soufriere is that of Cole et al. 2019, but those authors do not even mention viscosity in their paper. Please explain.

The correct inferred viscosity of 10^{11} Pa.s, the 11 was not in superscript in the original submission hence the misinterpretation by the reviewer, however, the sentence was removed (line 204 - 206).

Section 2 on Crisis Response. I have outlined above my major concern relating to unclear definition of roles and responsibilities. I add that this section includes a higher proportion of sentences whose English is poorly effective contributing to generate confusion.

In responding to your broader comments, we have substantially improved the awkward phrasing and considerable revisions have been made to the entire manuscript.

Line 223. The reported source of uncertainty (“imperfect understanding”) is only the epistemic component of uncertainty. There is another component which is not mentioned and can be equally or even more relevant. This is the aleatoric component of uncertainty, generated by the complex behavior of highly non-linear volcanic systems. Overlooking the aleatoric component of uncertainty may be dangerous when interpreting complex sequences of data, as it may lead to over-interpretations and poorly justified conclusions.

We have now significantly improved this and referred to a relevant paper that deals with uncertainty but also more clearly stated the particular relevance to how uncertainty in eruption forecasting was treated during the eruption (lines 260 – 266).

Line 232. Reference to “central value” is ambiguous, as there are many measures of the central tendency of a distribution. The median is cited two lines above, so I guess that’s the referred quantity? Please clarify.

Central value changed to median value (see line 271 and 273) and is the quantity referenced.

Line 234. Here it is said that the elicitation process led to a strong increase (from 20 to 60%) in the (median? Please clarify) probability of explosive activity in the morning of 8 April. What were the observations that led to such a dramatic change in the evaluations by the elicited experts?

Sentence revised to state median value. The observations that led to the change was the appearance of banded tremor (see lines 272– 273), which was interpreted as indicating an imminent explosive phase, with a source attributed to excitation of shallow gas and fluid pockets (lines 139 – 142).

Fig. 2 seems to show that no major changes occurred on April 8 (VT seismicity had increased in the previous days and does not seem to show anything major on 8 April). Low amplitude tremor is reported, is it the major cause of the jump in probability?

Correct, banded tremor reported and depicted in Figure 4, was the major cause of the jump in probability (see previous comment above).

Line 234: SVG (Saint Vincent and the Grenadine, I guess) is not previously introduced.

Revised and introduced in line 84.

Line 243 (and others similar): put a space between numbers and their dimensions.

Paragraph revised and sentence removed (lines 278 - 284), however, the information on visual observations in the early days of the explosive transition is now reported in lines 236 - 240.

Discussion section. This is all highly speculative, the kind of conceptual modeling any volcanologist extracts from sequences of data and observations, in this case with relatively limited data.

This main objective of the manuscript is to show how an observatory translates science into actionable forecasts and hazard assessments and how it communicates these with the public as part of an ongoing eruption response. This is something that is currently a gap in the literature. The Discussion section has been revised to reflect the broader implications of this eruption response, and how it might inform crisis response elsewhere, particularly for responding to eruptive transitions thereby widening the scope of the manuscript (see detailed response to main comment by Reviewer #1 above).

Reviewer #3:

Lindsay review of “The 2020 – 2021 eruption of La Soufrière volcano, St. Vincent: Monitoring and Scientific Response” by Joseph et al.

Overview

This paper presents an overview of the monitoring data during the 2020 – 2021 eruption of La Soufrière volcano in St. Vincent, and how the data, combined with prior knowledge and experience of the volcano, were used to inform decision-making, crisis communication and assessment of risk. This is a very interesting manuscript that provides new insights into a very recent eruption, and that highlights some of the challenges of responding to volcanic crises, especially in resource-constrained contexts. I think it will be of interest to a wide audience as it provides a comprehensive summary of the eruption, the monitoring data that were obtained, and the actions that were taken – including the development of an evolving

conceptual model. Overall, it is a well-written paper and my comments are generally quite minor. I understand that papers published by Nature Communications “represent important advances of significance to specialists within each field”. Although this manuscript doesn’t present advances as such, I am of the opinion it is extremely important to document such eruptions, eruption responses, and the link between data streams and evolving conceptual models. If required for publication in Nature Communications I am sure the manuscript could be tweaked so that its focus is more on the novel aspects.

See comments above in response to Reviewer #2 about the importance of thinking about science in context in the case of a volcanic eruption and we present a new and exciting dataset in this regard.

General comments and suggestions:

I found it unusual that the hazard zones in the hazard map were labelled in the key as red, orange, yellow and green, instead of an explanation of what the zones mean. That’s the equivalent of a key to a dotted area on a map just saying “dots” in the legend. Is this really the map that was used? It may be that the population on St Vincent are so familiar with the hazard zones that it is sufficient to just show the colours with no explanation – but I think for an international audience the original descriptions of the zones are required somewhere.

Agreed. Changes made to the label and caption of Figure 1 to provide a description of the zones.

I wonder if it would be worthwhile providing the actual results of your life-safety risk assessment and how these evolved during the crisis—and an explanation for how this affected decisions made to undertake field work? This would be of interest to other observatories and agencies I am sure.

Revised to include an example of the results of the life-safety risk assessment in the context of the initial period of vital network-strengthening in high-risk areas and the operational impact has been added (lines 457 - 459), additional supplementary figure now included to depict the results of the life-safety risk assessment.

Also – in the risk assessment section on page 23 it seems you really only describe in detail the expert elicitation related to the evolving eruptive processes (ie the second approach) – which is not a risk assessment in itself. Maybe make this section clearer by providing a bit more information about how the results of the expert elicitation fed into any actual risk assessments, including for example how (if at all) the expert elicitation that you describe here feeds into the life-safety risk assessment.

The title of the section has been revised (line 596) to reflect both the risk assessment and the evaluation of hazard via the expert elicitation for eruption scenarios. This section has been revised for clarity and to include a more detailed description of the life-safety risk assessment procedure and how the elicited eruption scenario probabilities feed into it (lines 607 – 615).

Specific comments:

Page 1 line 20 – “in writing the manuscript and was in charge.....”

Change made.

Page 1 contributions section – be consistent about how you present peoples’ initials. Some have full stops some don’t

Changes made.

Abstract – line 42 – add 2021 after April.

Abstract revised (lines 36 – 49).

Abstract – line 43, delete comma after forecasts. Change By to “In contrast”

Abstract revised (lines 36 – 49).

Abstract – line 45, add hyphen – resource-constrained

Change made (line 47).

Page 3 lines 50 and 51 – I got very confused with the two ones, superscripts. To avoid confusion, maybe remove the footnote and just define crisis science in the text.

Page 3 – definition in the foot note – this was confusing because the text said crisis science yet the footnote explained “science during crisis”. I would stick to the same wording if you are keeping the footnote – but suggest you get rid of it and explain in the text: “Crisis science is defined here as conducting.....”. Also – in the footnote you refer to this “report” – rather than “here” or this “paper”.

Change made (lines 59 – 64).

Page 3 – line 53 suggest a change to “.....and 300-600m deep4”

Change made (line 66).

Page 3 line 61 – space after 1971.

Sentence revised (line 76 - 79).

Page 3 line 66 – remove comma after 2020

Change made (line 82).

Page 4 line 82 – suggest deleting “also” before underpin.

Paragraph revised (line 95 – 97).

Page 4 line 83 – well-documented (add hyphen). Also - I suggest saying “We also reflect on....”

Changes made (lines 96).

Page 4 line 91 – should anglophone be with a small a?

Paragraph revised (lines 101 – 105).

Page 5 – lines 110 and 113 – I suggest adding 2021 after each mention of January

Changes made (line 118).

Page 6 lines 119-120. This first sentence is confusing. The seismicity associated with inter-volcanic episodes at La Soufrière is generally sparse, but interspersed by irregular, short-lived swarms. Is the sparse seismicity (whatever that looks like) really interspersed with irregular short lived swarms, or do the short lived swarms actually represent the sparse seismicity? Maybe rephrase.

Sentence deleted and paragraph revised (lines 128 – 130).

Page 6 line 121 – suggest adding a 2020 after December.

Change made (line 129).

Page 6 line 129 – There is no Fig 2A – I believe this should refer to 3A?

Correct. Paragraph revised (lines 134 -138).

Page 6 line 129 – add 2021 after March

Change made (line 136).

Page 6 line 130 - Figure 3 doesn't show hypocentres – rather epicentres. Change to epicentres

Paragraph revised (lines 133 -138).

Page 6 line 130 delete comma from after “swarm”

Paragraph revised (lines 133 -138).

Page 6 line 134 – add 2021 after April
Change made (line 136).

Page 7 line 143 – add “the” in “hours of the explosion”
Change made (line 151).

Page 7 line 145 should the reference here be to 4B?
Figure 4 was revised; the reference is now 4A (line 152).

Page 7 line 146 – there is no Fig 4C – double check what figure you are referring to
Change made to reference Fig 4B (line 161).

Page 7 line 150 and 151 – can these two sentences be combined into one? Also – delete comma after (Fig. 1)
Paragraph re-written (lines 166 – 172).

Page 7 line 155 – this sentence is confusing. “Subsequently no unambiguous deformation signal was detected from the SAR platforms”- that means that all subsequent deformation signals were ambiguous. Or do you mean – no deformation at all was observed?
Unambiguous deleted (line 170).

Page 7. Line 157. It is impossible to see 9 April on the figure. Add ticks that can be seen, and maybe label 9 April?
Figure 2 revised, labels added.

Page 7 line 162 – what do you mean by analysis of dome removal? Should that actually read dome volume?
Sentence comparing volume of erupted material removed over different time spans removed (lines 178 – 180).

Page 8 line 167 – add 2021 after January, and add comma after instruments
Changes made (lines 184).

Page 8 line 170 - change were to was. SO2 is singular
Change made (line 186).

Page 8 line 173 traverse of..... what? A crater lake? In the sea off the coast under the plume?
Sentence revised (see line 191).

Page 8 line 180 – delete brackets, this makes the sentence easier to understand.
Paragraph revised (lines 197 – 200).

Page 9 – line 200 – add 2021 after March.
Change made (line 227).

Page 10 – line 214 delete comma after Fig. 1.
Change made (line 245).

Page 10 line 228 – delete comma after March 2021
Change made (line 267).

Page 11 line 232 – what do you mean by “central value”?
Changed to median value (line 271).

Page 11 line 243 – Maybe add a sentence about what happened between 12 April and 22 April? At present you jump straight to the end of the eruption without saying what happened in between.
This paragraph revised and moved to Section 1.5 and now includes additional information covering the period 12 to 22 April (see lines 236 – 240).

Page 13 line 291 – Consider changing “construed” to a better word. Do you mean presented? Otherwise you are effectively saying “interpretations need to be interpreted in the context.....”
Construed replaced by presented (line 340).

Page 13 lines 293-295 This sentence is a bit clumsy. “Aims” don’t require rapid response – “achieving aims” might.... The second part of the sentence is also unclear. “Rapid response” of monitoring data doesn’t make sense, for example. Maybe – “In our experience, achieving these aims requires rapid collection of monitoring data that are then analysed, interpreted and communicated in light of the experience accrued from past activity at the volcano as well as long-term engagement with stakeholders and communities on the ground. However, you have to make sure that this isn’t just a repeat of what is in the previous sentence, at the moment the way you have written it reads as saying the same thing in different words.
Paragraph revised (lines 337 – 347).

Page 14 page 312 – It is unclear to me why this process (injection of gas rich magma) would lead to low rates of seismicity. Maybe spell that out.
The initial low rate of seismicity was interpreted to be a consequence of a ductile, well-connected magma ascent pathway. (see lines 364 – 365).

Also – in line 310 – what is the gas rich magma being injected into? Do you mean – injected into a shallow reservoir?
This is correct (see line 360 - 362).

Page 14 – line 313, I suggest changing “being largely focused on” to “expanded to infer the presence of”
Page 14 line 314 – maybe “degassed magma that had remained within the conduit following the 1979 eruption”
Sentence revised (see lines 376 – 379).

Page 14 line 317 – add a “had” before reinforced
Change made (line 380).

Page 15 line 323 – add commas after region and after crust.
Changes made (lines 369).

Page 15 line 323 – “higher impact” eruption is vague. Do you mean larger eruption? More explosive activity? More voluminous activity? Longer duration activity?
Sentence revised: explosive activity (see lines 370– 372)

Page 15 line 328 add 2021 after April, then changed halted to followed
Paragraph revised (see lines 384 - 390).

Page 15 line 330 and 331. It sounds like you are talking about hypocentres but you say epicentres. Just

check that you actually do mean epicentres (and maybe refer to the figure that shows the change in epicentral patterns if that is indeed what you are referring to)
Hypocentres changed to epicentres. Figure 3 now referenced (line 385 - 387).

Page 15 line 340 VT events (plural)
Change made (line 399).

Page 15 line 344 – maybe remind us here what was happening with gas.
Sentence revised (see line 399 - 401).

Page 16 line 346 – Refer to the figure after deflation
Change made (line 404).

Page 16 line 359 – conceptual model rather than models?
Change made (line 405).

Page 16 Line 362 – the signals themselves were not uncertain. Maybe change to unclear? Inconclusive?
Sentence revised (see lines 421 – 423).

Page 16 line 362- 363 maybe change to “(iii) once explosions started the extent to which they might continue was unknown, given the range in size and intensity of previous explosions”
Paragraph revised (see lines 421 – 439).

Page 17 – line 372 this sentence doesn’t make sense. Maybe delete “understanding”?
Sentence revised (see lines 444 - 445).

Page 17 lines 373 onwards – maybe provide an actual example of a process and a link with monitoring data?
Sentence deleted and example of improvements in data volume and accuracy linked with the examples of the improved sensitivity of the GPS network (lines 444 – 445), as well as improvements in seismic analysis and measurement of SO₂ flux was the most salient information to feed into changed views on explosion likelihood (lines 449 – 453).

Page 17 line 386 – delete hyphen from field-practices
Change made (line 461).

Page 17 line 388 “Than is acceptable in other...”
Sentence revised and now gives a quantification of the risk (lines 457- 459) with additional supplementary figure.

Page 18 line 401 – should the IAVCEI reference have a superscript? Maybe give the title and then a superscript to the full reference in the reference list
Sentence revised lines 465 – 466 and IAVCEI reference is now listed as citation 43.

Page 18 line 409 – Perhaps the best indicators (plural – you present more than one) also change was to were
Sentence revised (lines 471 – 475).

Page 18 line 410 – the way this is written could be interpreted to mean the authorities acted decisively by raising alert levels. Maybe tweak so it is clearer?
Sentence revised (see lines 471 - 475).

Page 19 – line 415 – change consequently to consequent
Section revised (lines 511 - 523).

Page 19 line 418 – maybe say “the monitoring network” rather than just the network. Also perhaps change to: The rapid strengthening of the monitoring network, the continuous stream of communication between scientists and authorities, and the use of a common framework to anticipate changing behaviour are key factors. (also – do you mean common framework – or conceptual model?)
Section revised (see lines 517 – 523).

Page 19 – line 423 – perhaps start this sentence with “In the case of the 2020-2021 eruption of La Soufriere in St Vincent, we propose that.....”
Section revised and change made (lines 515 – 515).

Page 19 line 425 – maybe delete “and work”
Section revised (see lines 511 – 523).

Page 19 line 427 – all-important conceptual models? (otherwise you could be talking about computer or numerical process models)
Paragraph revised and the term ‘flexible conceptual model’ is now included (lines 486 487).

Page 19 line 429 – maybe modify to something like “.....heightened activity is more readily translated into actions by local emergency managers, and in turn, the populace at risk” (I don’t think health and safety officials is the right phrase here)
Paragraph revised (lines 491 – 493).

Methods

Page 20 Line 441 – delete comma after generated
Change made (line 530).

Page 23 – as mentioned above, it would be interesting to see how the results of the life safety risk assessment evolved during the crisis. Also – in this section it seems you really only describe in detail the expert elicitation related to the evolving eruptive processes (ie the second approach) – which is not a risk assessment in itself. Maybe make this section clearer by providing a bit more information about the life-safety risk assessment – including for example how (if at all) the expert elicitation that you describe here feeds into it. And, how the results fed into any actual risk assessments.
Section revised as noted above (lines 596 – 614) and a new Supplementary figure is included.

Page 24 line 536 – delete comma after population
Paragraph revised (see line 638 - 643).

Page 24 line 538 and elsewhere – what do you mean by gender-sensitive materials? Maybe explain this somewhere?
Sentence rewritten to clarify (see lines 647 - 650). This was an incorrect phrase, the correct phrase is gender sensitive community sessions.

Acknowledgements – line 586 delete “eruption” after soufriere.
Change made (line 715).

Also – field work for JB and PDC – unclear who PDC is given Paul Cole doesn’t have a middle name in the list of authors

Middle initial for Paul Cole deleted.

Figures

Figure 1

– See comment about the strange legend. What do the hazard zones actually show?

Caption revised to include the sentence: Hazard zones illustrate the potential for ground-based volcanic impacts such as pyroclastic flows and surges, tephra fall, ash fall and lahars that may impact the defined areas

– The red triangles are hard to see against the red background. Maybe give them a black outline?
(probably good for the green circles too)

– Maybe change the first sentence in the caption to: “Volcanic hazard map of LS showing the monitoring network” or – “Volcano monitoring network for La Soufriere, St. Vincent plotted on the volcanic hazard map”

Caption revised as suggested.

Figure 2

- C/Stot – needs explaining in the caption. Also use the subscript for tot.

Changes made in caption.

- Improve the x axis labels – how can a single tick represent a whole month? Are they supposed to indicate the first day of each month? If that is the case, then that should be the label. I suggest also including some lower-level ticks, and labelling / annotating more key dates (for example start and end of explosive phase)

X axis revised to include suggestions.

Figure 3

- You could probably be a bit more specific in B and C and say these are epicentral plots

Figure 4

- The minor ticks on the x axis are not visible – maybe make them bigger? Otherwise it is very hard to estimate what date falls where in the plots.

- There is no C in this figure yet you refer to it in the text.

Figure 3 has been revised.

Figure 6

- Fix the subscripts in the figure? And the caption

Figure and caption revised.

Figure 10

- I found the dots in this figure confusing. Maybe in the caption provide an example of “how to read” the table, by explaining what the dots mean in the context of one single entry?

Figure 10 revised.

Supplementary material – It would be good to have more detail in the captions about what we are actually looking at in these two figures. What was the first flyer used for? Who was the audience? In the second figure – who is in all the figures? The journal will likely require written permission from everyone in the photographs in order to publish them. What are we looking at exactly? Are we looking at screen shots of radio interviews? You tube videos? Please explain in the caption.

This addition to the Supplementary material was removed from the revised submission.

REVIEWERS' COMMENTS

Reviewer #1 (Remarks to the Author):

I have read the revised version of the manuscript, and I think it is much improved over the original submission. Specifically it addressed my main concern on the lack of broader impact, but this has been fixed with a new section in the discussion about anticipating effusive/explosive transitions. Overall I don't see any need for further revisions.

Reviewer #2 (Remarks to the Author):

I have read the responses by the authors to my comments, and the revised manuscript. Although I appreciate the authors' efforts, my conclusion remains that this manuscript is 1) out of scope for this journal as it does not address any "important advances of significance to specialists", nor it aims at doing so; 2) not original or novel, as it tells a story which is not dissimilar from that of other cases of claimed success in managing volcanic risk at poorly (or relatively poorly) monitored volcanoes. As from my previous conclusion, as a reader I would ask myself why this particular account is given the status of a highly impacting journal publication, instead of being more conveniently hosted in one specialized journal as for other similar accounts.

The arguments above do not refer to scientific or technical flaws but to convenience and appropriateness for the manuscript to become a Nature Communications paper. As such, any decision thereby is more appropriately made by the journal Editors.

Reviewer #3 (Remarks to the Author):

I have had the chance to review the revised manuscript and I think you have done a good job addressing all the reviewers' concerns. Although it is a sometimes a challenge to justify publications that report on an eruption crisis rather than a new an exciting scientific discovery, such documentation of eruptions is extremely valuable. In your revisions and response to reviewers' comments you have made efforts to more clearly point out the scientific contribution your work is making. Well done!

The only think I couldn't find in the revised manuscript were the Figure captions. It may be that there are there and I just couldn't see them - but maybe double check.